# *Self-Aug*: QUERY AND ENTROPY ADAPTIVE DECODING FOR LARGE VISION-LANGUAGE MODELS

**Eun Woo Im**[1]*, **Muhammad Kashif Ali**[2], **Vivek Gupta**[1]
[1]Arizona State University    [2]Friedrich-Alexander-Universität Erlangen-Nürnberg
{eunwooim, vgupt140}@asu.edu, kashif.m.ali@fau.de

## ABSTRACT

Large Vision-Language Models (LVLMs) have demonstrated remarkable multimodal capabilities, but they inherit the tendency to hallucinate from their underlying language models. While visual contrastive decoding has been proposed to mitigate this issue, existing methods often apply generic visual augmentations that disregard the specific context provided by the text query, limiting their effectiveness. This study introduces a novel training-free decoding strategy that addresses these limitations, featuring two key contributions. First, a self-augmentation prompting strategy that leverages the intrinsic knowledge of the model to dynamically align semantics between the query and the visual augmentation. Second, an adaptive thresholding algorithm that adaptively adjusts next token candidate size based on the output sparsity, utilizing full information from the logit distribution. Extensive experiments across five LVLMs and seven benchmarks demonstrate that the proposed decoding significantly enhances factual consistency compared to state-of-the-art decoding methods. This work highlights the importance of integrating query-dependent augmentation and entropy-aware decoding for improving effective generation of LVLMs.

## 1 INTRODUCTION

Large Language Models (LLMs) have achieved remarkable success in language comprehension, generation, and reasoning (Brown et al., 2020; Google, 2023; Touvron et al., 2023; Chiang et al., 2023; OpenAI, 2023). By integrating visual encoding and projection, Large Vision-Language Models (LVLMs) have extended these capabilities to multimodal applications such as visual perception and planning (Li et al., 2022; Yu et al., 2022; Li et al., 2023a; Maaz et al., 2023; Ye et al., 2023; Zhang et al., 2023a; Zhu et al., 2023; Huang et al., 2023). Despite their impressive performance, LVLMs inherit critical limitations from their foundational language models. One of the most significant issues is *hallucination*, a phenomenon of generating plausible but factually incorrect or nonsensical outputs. This behavior is largely a byproduct of the auto-regressive training objective of the model, a process that incentivizes a reliance on spurious correlations over a precise understanding of underlying facts by maximizing token likelihood based on surface-level statistical patterns (Bender & Koller, 2020).

Advanced decoding methods can significantly enhance the factual consistency by shaping how token sequences are selected from output distributions at each generation step (Van der Poel et al., 2022; Favero et al., 2024). A prominent decoding strategy to reduce hallucination effect is Contrastive Decoding (CD) (Li et al., 2023c), a technique that improves factuality by contrasting the outputs of an expert model with those of a weaker, *amateur* counterpart (Zhang et al., 2023b; Chuang et al., 2023). Motivated by this principle, Visual Contrastive Decoding (VCD) (Leng et al., 2024) was introduced to improve the general perceptual capabilities of LVLMs by contrasting standard output with an amateur logit generated from an input image degraded by random noise. Subsequent research in VCD has primarily focused on determining which visual modifications or hidden states with experimental heuristics can maximize the sample variance while maintaining the semantics (Li et al., 2023b; Huang et al., 2024). However, these methods often overlook the critical role of the *input text query*, which specifies which aspects of an image are relevant to the user request. For instance, asking to identify an object in the image and solving a handwritten math problem require entirely different capabilities

---

*Corresponding author. Project page: **https://eunwooim.github.io/selfaug**

and reasoning from the LVLM. While VACoDe (Kim et al., 2024b) addressed this by estimating the divergence between logit distributions among the predefined visual augmentation set at the first generation step in a brute-force manner, there are two fundamental limitations. First, first-token divergence is an empirical measure that does not always assure a favorable augmentation choice for the entire generation sequence. Second, its dependence on a single token renders it suitable for short, multiple-choice style answers but fundamentally limits its effectiveness for complex tasks requiring open-ended generation and multi-step reasoning.

Moreover, a challenge in contrastive decoding arises from the subtraction of the amateur logit from the expert logit (Li et al., 2023c). This operation can cause undesired effects that amplify the scores of certain tokens; if the amateur model produces a negative logit value, it will have its final score erroneously increased (Lyu et al., 2024). To mitigate this amplification effect, existing methods (Jin et al., 2024) truncate the vocabulary set based on a threshold set proportionally to the maximum value of the expert logit distribution. However, while this approach is effective at penalizing false positives, its reliance on a single data point (*i.e.*, the maximum logit) hinders it from utilizing the rich information encoded in the full logit distribution, such as *model confidence*.

These aforementioned limitations lead us to two main research questions. (1) How can the semantic intent of a text query guide the selection of a visual augmentation[1] to elicit a maximally informative discrepancy for contrastive decoding? (2) Is there a correlation between a predictive confidence of the model and the plausibility of its next-token candidates? To address these questions, this study introduces *Self-Aug*, a novel decoding strategy that adaptively select which visual augmentation is best suited to be *contextually relevant*. Unlike prior works (Kim et al., 2024b), *Self-Aug* utilizes the intrinsic model knowledge to determine an optimal visual modification out of the box. Furthermore, we introduce an improved thresholding algorithm, Sparsity Adaptive Truncation (SAT), to overcome the limitations of existing plausibility constraints. Where prior methods often fail to utilize full information from the logit, SAT dynamically determines a threshold by utilizing the entire logit distribution as a proxy for the confidence of the output. The proposed method integrates seamlessly into any LVLM without requiring any architectural modifications or additional training. Extensive experiments and analysis verify that the proposed methods significantly enhance factual consistency and reduce hallucinations across multiple models and benchmarks. The contributions of this study are summarized as follows:

1. This work introduces *Self-Aug*, a prompting strategy that leverages parametric knowledge of the model to select a visual augmentation that is semantically relevant to the textual query, thereby extracting a more informative discrepancy.

2. The proposed SAT improves the existing adaptive plausibility constraint by leveraging the entropy of the expert logit and dynamically sets a threshold of token implausibilities.

3. Extensive experiments validate the effectiveness of the proposed method across five LVLMs and seven benchmarks. The results demonstrate that *Self-Aug* significantly reduces hallucinations while amplifying the relevance and informativeness in the response.

## 2 PRELIMINARIES

**Auto-regressive Generation of LVLMs** Suppose that $f_\theta$ is an LVLM (Gong et al., 2023; Maaz et al., 2023; Li et al., 2025a), parameterized by $\theta$. The model operates on a vocabulary set $\mathcal{V}$, and the set of all possible token sequences can be denoted by its Kleene closure, $\mathcal{V}^* = \bigcup_{i \geq 0} \mathcal{V}^i$, where $i$ indicates the timestamp of the LVLM output. The function $f_\theta : \mathcal{V}^* \times \mathbb{R}^{h \times w \times 3} \to \mathcal{V}^*$ auto-regressively generates a response from a given text query $x \in \mathcal{V}^*$ and a visual input $v \in \mathbb{R}^{h \times w \times 3}$. At each timestep $t$, the LVLM computes a logit distribution over the vocabulary for the next token $y_t$, conditioned on the inputs $(x, v)$ and the sequence of previously generated tokens $y_{<t}$. This yields the probability distribution over the next token:

$$p_\theta(y_t | v, x, y_{<t}) \propto \exp\left(\text{logit}_\theta(y_t | v, x, y_{<t})\right). \tag{1}$$

The next token is then selected from this distribution according to a chosen decoding method. Decoding methods are broadly categorized into two families: deterministic search, including greedy

---

[1]Throughout this paper, we use the terms "augment" and "modify" interchangeably, consistent with their usage in the related literature.

and beam search (Graves, 2012), and stochastic sampling, such as top-k, Nucleus (Holtzman et al., 2019), Mirostat (Basu et al., 2020), and typical (Meister et al., 2023) sampling.

**Hallucination**   Ideally, the generated response $y$ should be factually accurate, relevant to the query $x$, and faithful to the visual content $v$. However, current LVLMs often fail to meet these criteria, suffering from a critical issue known as hallucination (Rohrbach et al., 2018). This phenomenon stems from multiple reasons, including imperfect learning and decoding (Ji et al., 2023), misalignment of vision and language modalities (Tong et al., 2024), and failure of understanding the context (Daunhawer et al., 2021). To address this issue, recent studies have suggested scaling the input image resolution (Liu et al., 2024b; Chen et al., 2024b), combining another inductive bias of visual encoders (Li et al., 2025b), post-hoc rectifying (Zhou et al., 2023), self-correction after generation (Yin et al., 2024), and advanced decoding methods (Shi et al., 2024; Favero et al., 2024). Among those approaches, decoding-based methods are particularly promising since they enable real-time control, do not require additional training, and are compatible with other hallucination mitigation strategies.

**Contrastive Decoding**   CD (Li et al., 2023c) tackled hallucination problems in the NLP domain by contrasting the predictions of two different language models with different capacities. VCD (Leng et al., 2024) extended the idea of CD with vision modality and introduced the contrastive counterpart $v'$ by degrading visual content with random noise to $v$. It sequentially treats the logit from $v'$ as an output of the amateur model, sampling the next token from:

$$p_{\text{CD}}(y|v, v', x) = \text{softmax}\left((1 + \alpha) \cdot \text{logit}_\theta(y|v, x) - \alpha \cdot \text{logit}_{\theta'}(y|v', x)\right), \qquad (2)$$

where $\alpha$ denotes an amplification parameter. Recent studies have focused on curating a better selection of the degradation to achieve maximal differentiation while preserving semantic integrity. For instance, cropping the patch which is likely to cause hallucinations (Chen et al., 2024a), caption substitute (Kim et al., 2024a), and visualization of the textual output (Park et al., 2025). While most VCD methods rely on a shared underlying principle of query-agnostic input modifications, VACoDe (Kim et al., 2024b) has introduced a dynamic visual augmentation strategy. This approach attempts to be query-aware by using $L_2$ distance between the expert and amateur logit distribution at the first token generation as a score function to select an augmentation.

However, this reliance on a first-generated token has fundamental limitations. The overall semantics of a task are not universally guaranteed to be reflected by first-token divergence, which is an empirical proxy. One example of failure is when two logits have the same argmax but are distinct in terms of overall entropy. In this case, the query could not be invalidated because, despite the distortion, the model could still answer correctly, although the divergence remained large. This may be effective for short-answer and multiple-choice questions, but it can be ineffective for other scenarios including open-ended questions and multi-step reasoning.

## 3 *Self-Aug*: SELF-AUGMENTED VISUAL CONTRASTIVE DECODING

To address the preceding limitations, we introduce *Self-Aug*, a decoding method that identifies a query-specific visual augmentation to apply for visual contrastive decoding by utilizing the rich knowledge base of LVLM (Li et al., 2024). Unlike prior methods that rely on simple heuristics, *Self-Aug* leverages the world knowledge and common sense embedded in the LVLM to achieve a *semantic alignment* between the query and the selected augmentation. This approach enables the model to reason the *underlying intent* of a query and make a choice that elicits a more targeted and informative discrepancy. Alg. 1 and Fig. 1 outline the proposed method.

### 3.1 SELF-AUGMENTATION SELECTION

**SAS Prompting**   Self-Augmentation Selection (SAS) is a concept of meta-level classification task that aims to employ parametric knowledge of the LVLM to dynamically select the best *task-optimal* visual augmentation on the fly that amplifies the output divergence. This is achieved through a structured SAS Prompt $\mathcal{P}$, which comprises three key components. First, the prompt contains explicit definitions of each visual augmentation and corresponding effects, providing the model with the necessary operational knowledge. Second, to minimize the risk of post hoc rationalization, the prompt is structured to elicit reasoning before the final selection is made (Zelikman et al., 2024). Finally,

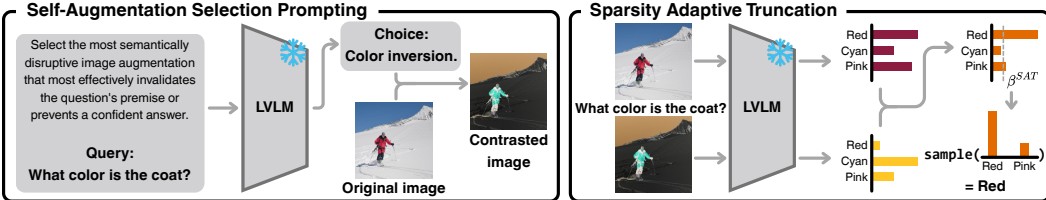

Figure 1: Overview of the proposed augmentation choice process and sparsity adaptive truncation.

inspired by few-shot learning techniques (Brown et al., 2020; Patel et al., 2024), in-context learning (ICL) examples are included in the prompt $\mathcal{P}$ to further condition the contextual knowledge (Alayrac et al., 2022). The textual output is then processed by a parsing function $g(\cdot) : \mathcal{V}^* \to \mathcal{V}^* \times \mathcal{V}^*$, which separates the reasoning trace $r$ and final augmentation choice $c$. The contrasted image is obtained by feeding the $v$ and the final choice $c$ to a predefined visual augmentation function $\mathcal{A}$.

$$(r, c) = g(f_\theta(\mathcal{P}, x)), \quad v' = \mathcal{A}(c, v). \tag{3}$$

Subsequently, contrasted logit distribution is calculated from expert logit $l = \text{logit}_\theta(y_t | v, x, y_{<t})$ and amateur logit $l' = \text{logit}_\theta(y_t | \mathcal{A}(c, v), x, y_{<t})$. The augmentation set is defined with random crop, random mask, noise addition, color inversion, horizontal flip, and vertical flip. Note that the generation configuration is set to greedy decoding for SAS Prompt $\mathcal{P}$ to ensure computational efficiency, determinism, and reproducibility. While further optimized prompting techniques (Manakul et al., 2023) and multiple combinations of different augmentations can be deployed, we limit the scope to two prompting features and the aforementioned augmentation set in this work. Full prompt is referred to the Appendix B.1.

## 3.2 RETHINKING ADAPTIVE PLAUSIBILITY CONSTRAINT

CD-based methods encourage the generation of implausible tokens since the output distribution from contrasted visual input $v'$ still involves the underlying semantics of $v$ (Li et al., 2023c). This can cause the two distributions to not cooperate properly, resulting in the reward of undesirable tokens. Adaptive Plausibility Constraint (APC) (Li et al., 2023c; Leng et al., 2024) addresses this challenge with a controllable hyperparameter $\beta \in [0, 1]$, setting a threshold proportional to the logarithm of the maximum probability of the new token, formulated as:

$$\mathcal{V}_{\text{APC}} = \{y_t \in \mathcal{V} \mid p_\theta(y_t | v, x, y_{<t}) \geq \beta \cdot \max_{w \in \mathcal{V}} p_\theta(w | v, x, y_{<t})\}. \tag{4}$$

However, since this thresholding mechanism is based solely on the maximum logit value and the meaning of a logit value is *relative* to the other logits in the distribution, it is a *confidence-agnostic filter*. Although this approach penalizes false positives by truncating the sample space, it becomes unreliable in low-confidence states, when the risk of discarding the correct token from the candidate set is high (Guo et al., 2017; Wenkel et al., 2021). This deficiency arises because APC disregards the rich signal encoded in the full output distribution. The *entropy* of logit distribution provides a more robust and holistic measure of model uncertainty which can be leveraged for a more effective filtering of the candidate set.

Model uncertainty, characterized by the value of output entropy, is a recognized correlate of model errors (Manakul et al., 2023). When the logit distribution is highly entropic, a more lenient threshold is required to create a sufficiently inclusive candidate set and avoid erroneously discarding the context-relevant tokens. Conversely, in low-entropy scenarios where the model is confident (Tornetta, 2021) with a sparse output distribution, a more restrictive threshold is required to retain pivotal tokens with high probability and to refine the candidate set by taking over the probability mass from filtered tokens (Li et al., 2023c). This *inverse proportionality* heuristic improves generation fidelity by minimizing the risk of sampling erroneous, low-probability tokens on the tail of the distribution.

To enable the *confidence-aware* thresholding, the proposed method extends APC to SAT, a method that dynamically adjusts the plausibility constraint based on the *sparsity* of the output distribution. The method leverages the principle that a sparsity is inversely related to its uncertainty, quantified by Shannon Entropy $H : \mathbb{R}^d \to [0, \log_2 d]$ (Shannon, 1948), which maps a probability distribution

---

**Algorithm 1** *Self-Aug*: Self-Augment Visual Contrastive Decoding

---

**Require:** input image $v$, text query $x$, LVLM $f_\theta$, augmentation function $\mathcal{A}$, SAS Prompt $\mathcal{P}$, vocabulary set $\mathcal{V}$, hyperparameter $\alpha$.

1: $c \leftarrow f_\theta(\mathcal{P}, x)$             ▷ Identify augmentation $c$ from given $x$
2: $t \leftarrow 0$                    ▷ Initiate $t$
3: **while** $t < T$ **do**
4:   $l \leftarrow \text{logit}_\theta(y_t|v, x, y_{<t})$          ▷ Set expert logit $l$
5:   $l' \leftarrow \text{logit}_\theta(y_t|\mathcal{A}(c, v), x, y_{<t})$       ▷ Set amateur logit $l'$
6:   $l_{\text{CD}} \leftarrow (1 + \alpha) \cdot l - \alpha \cdot l'$        ▷ Set contrasted logit
7:   $\beta_t^{\text{SAT}} \leftarrow H_{\text{decay}}(\text{softmax}(l))$       ▷ Set SAT parameter $\beta_t$ from Eq. 5
8:   $\mathcal{V}_{\text{SAT}} \leftarrow \{y_t \in \mathcal{V} \mid p_\theta(y_t|v, x, y_{<t}) \geq \beta_t^{\text{SAT}} \cdot \max_{w' \in \mathcal{V}} p_\theta(w'|v, x, y_{<t})\}$   ▷ Set threshold
9:   $l_{\text{CD}}[i] \leftarrow -\infty$   **for all** $i \notin \mathcal{V}_{\text{SAT}}$     ▷ Apply vocabulary truncation
10:   $y_t \sim \text{softmax}(l_{\text{CD}})$           ▷ Token sample
11:   $t \leftarrow t + 1$
12: **end while**
13: **return** $\{y_0, ..., y_{T-1}\}$

---

$p$ over its dimension $d$ to its uncertainty, calculated as $H(p) = -\sum_{i=0}^{|\mathcal{V}|-1} p_i \log_2 p_i$. To implement an inversely proportional relationship where higher entropy yields a smaller threshold, a decayed entropy function $H_{\text{decay}} : \mathbb{R}^{|\mathcal{V}|} \to (0, 0.5]$ is formulated to compute the threshold value:

$$H_{\text{decay}}(p) = \sigma\left(-\gamma \sum_{i=0}^{|\mathcal{V}|-1} p_i \log_2 p_i\right), \tag{5}$$

where $\sigma$ and $\gamma < 0$ denote a sigmoid function and a scaling parameter, respectively. The choice of a sigmoidal decay is deliberate, as other decaying functions, such as exponential or polynomial (Provencher, 1976; Borichev & Tomilov, 2010), could be potentially considered, but they lack the versatility of a sigmoid. The curve of the sigmoid function is naturally bounded to $(0, 1)$, and its lower plateau creates a stable, consistent threshold for low confidence distributions, and precise controllability over the single steepness parameter $\gamma$ of the decay for mid-range entropy. Furthermore, by ensuring the threshold remains strictly less than 1, SAT prevents the candidate set from collapsing to a single token, guaranteeing that the decoding process remains distinct from greedy decoding. The proposed SAT method introduces a dynamic threshold $\beta_t^{\text{SAT}}$, which is calculated by incorporating the entropy of the logit distribution: $\beta_t^{\text{SAT}} = H_{\text{decay}}(\text{softmax}(\text{logit}_\theta(y_t|v, x, y_{<t})))$. The next-token candidate set, $\mathcal{V}_{\text{SAT}}$, is then constructed by filtering the vocabulary set with this adaptive threshold: $\mathcal{V}_{\text{SAT}} = \{y_t \in \mathcal{V} \mid p_\theta(y_t|v, x, y_{<t}) \geq \beta_t^{\text{SAT}} \cdot \max_{w \in \mathcal{V}} p_\theta(w|v, x, y_{<t})\}$. To exclude the implausible tokens, $-\infty$ is assigned to logit elements which are not involved in $\mathcal{V}_{\text{SAT}}$. Finally, the contrasted probability distribution is obtained by combining Equations 2 to 5:

$$l_{\text{CD}}(y_t|v, x, y_{<t}) = \begin{cases} (1 + \alpha) \cdot l - \alpha \cdot l', & \text{if } y_t \in \mathcal{V}_{\text{SAT}} \\ -\infty, & \text{otherwise.} \end{cases} \tag{6}$$

The token at timestamp $t$ is sampled from $p_{\text{CD}}(y_t|v, x, y_{<t}) = \text{softmax}(l_{\text{CD}})$.

## 4 EXPERIMENTS

### 4.1 EXPERIMENTAL SETTINGS

**Benchmark and Model Selection** Following standard practices in the literature (Leng et al., 2024; Kim et al., 2024b), three foundation model families are selected to evaluate the effectiveness of *Self-Aug*: LLaVA-1.5-7B/13B (Liu et al., 2024a), Qwen-VL-7B (Bai et al., 2023), InstructBLIP-7B (Dai et al., 2023) with vicuna-v1.1 (Chiang et al., 2023), and Qwen3-VL-8B (Bai et al., 2025). The evaluations are divided into two categories: discriminative and generative benchmarks. Discriminative benchmarks assess the factuality of visual recognition in the form of binary or multiple choice questions, while generative benchmarks evaluate broader capabilities by requiring open-ended responses and judge with proprietary models (Zheng et al., 2023; Gu et al., 2024; Ali et al., 2025). The selected discriminative benchmarks include POPE (Li et al., 2023d) constructed on MSCOCO (Lin

Table 1: Discriminative benchmark results on MME (Fu et al., 2024), MMVP (Tong et al., 2024), and POPE (Li et al., 2023d) constructed on COCO (Lin et al., 2014), and A-OKVQA (Schwenk et al., 2022). Avg. Δ denotes averaged gain against Multinomial sampling across benchmarks.

| Model | Method | POPE-MSCOCO Acc.↑ | F1↑ | POPE-AOKVQA Acc.↑ | F1↑ | MME-P↑ | MMVP↑ | Avg. Δ |
|---|---|---|---|---|---|---|---|---|
| LLaVA-1.5-7B | Multinomial | $82.07_{\pm1.83}$ | $80.48_{\pm1.66}$ | $79.81_{\pm4.12}$ | $79.86_{\pm3.26}$ | $1278.42_{\pm30.30}$ | $32.40_{\pm4.73}$ | - |
| | VCD | $83.66_{\pm1.97}$ | $82.55_{\pm1.76}$ | $80.51_{\pm4.49}$ | $81.11_{\pm3.56}$ | $1323.67_{\pm20.84}$ | $34.00_{\pm3.89}$ | +10.86% |
| | VACoDe | $\mathbf{84.29}_{\pm2.41}$ | $\mathbf{83.59}_{\pm2.11}$ | $80.86_{\pm4.97}$ | $81.87_{\pm3.90}$ | $1372.50_{\pm13.78}$ | $\mathbf{36.67}_{\pm2.87}$ | +9.52% |
| | *Self-Aug* | $82.93_{\pm1.77}$ | $83.57_{\pm1.63}$ | $\mathbf{82.80}_{\pm4.75}$ | $\mathbf{83.20}_{\pm3.86}$ | $\mathbf{1431.30}_{\pm13.87}$ | $36.00_{\pm3.09}$ | **+14.32%** |
| LLaVA-1.5-13B | Multinomial | $83.86_{\pm1.51}$ | $81.02_{\pm1.33}$ | $80.97_{\pm3.51}$ | $80.79_{\pm2.84}$ | $1351.69_{\pm30.30}$ | $31.60_{\pm4.82}$ | - |
| | VCD | $83.86_{\pm1.72}$ | $82.68_{\pm1.53}$ | $81.93_{\pm3.65}$ | $82.16_{\pm2.94}$ | $1372.77_{\pm30.54}$ | $31.60_{\pm4.81}$ | +6.33% |
| | VACoDe | $84.86_{\pm1.90}$ | $\mathbf{84.17}_{\pm1.68}$ | $82.34_{\pm3.90}$ | $83.08_{\pm3.02}$ | $1434.09_{\pm12.79}$ | $32.13_{\pm3.25}$ | +8.03% |
| | *Self-Aug* | $\mathbf{85.37}_{\pm1.42}$ | $83.96_{\pm1.32}$ | $\mathbf{84.25}_{\pm3.64}$ | $\mathbf{84.13}_{\pm3.04}$ | $\mathbf{1462.18}_{\pm18.21}$ | $\mathbf{34.80}_{\pm1.19}$ | **+11.59%** |
| Qwen-VL | Multinomial | $75.72_{\pm0.79}$ | $72.07_{\pm0.85}$ | $76.64_{\pm2.50}$ | $74.68_{\pm2.16}$ | $1311.79_{\pm23.42}$ | $17.33_{\pm2.54}$ | - |
| | VCD | $77.98_{\pm0.79}$ | $75.42_{\pm0.77}$ | $78.85_{\pm2.62}$ | $77.73_{\pm2.77}$ | $1415.12_{\pm21.31}$ | $21.33_{\pm2.62}$ | +5.05% |
| | VACoDe | $\mathbf{78.35}_{\pm0.93}$ | $\mathbf{76.08}_{\pm0.86}$ | $\mathbf{78.98}_{\pm3.07}$ | $\mathbf{78.02}_{\pm2.77}$ | $1412.43_{\pm10.27}$ | $22.13_{\pm3.75}$ | **+7.49%** |
| | *Self-Aug* | $77.58_{\pm0.65}$ | $74.71_{\pm0.64}$ | $78.23_{\pm2.69}$ | $76.78_{\pm2.36}$ | $\mathbf{1442.36}_{\pm11.87}$ | $\mathbf{26.67}_{\pm1.63}$ | +6.69% |
| InstructBLIP | Multinomial | $68.70_{\pm1.74}$ | $69.34_{\pm1.42}$ | $65.52_{\pm3.00}$ | $68.36_{\pm1.91}$ | $973.66_{\pm41.81}$ | $19.20_{\pm1.52}$ | - |
| | VCD | $71.99_{\pm1.27}$ | $72.77_{\pm1.11}$ | $69.26_{\pm3.03}$ | $72.23_{\pm2.03}$ | $1079.39_{\pm46.30}$ | $18.93_{\pm2.77}$ | +12.33% |
| | VACoDe | $73.29_{\pm1.50}$ | $74.26_{\pm1.17}$ | $70.01_{\pm3.28}$ | $73.40_{\pm2.21}$ | $1090.88_{\pm33.01}$ | $\mathbf{21.87}_{\pm1.10}$ | +10.98% |
| | *Self-Aug* | $\mathbf{82.86}_{\pm1.94}$ | $\mathbf{82.34}_{\pm1.62}$ | $\mathbf{72.09}_{\pm3.76}$ | $\mathbf{75.37}_{\pm2.51}$ | $\mathbf{1198.53}_{\pm17.95}$ | $16.13_{\pm3.18}$ | **+18.78%** |
| Qwen3-VL-8B | Multinomial | $88.59_{\pm0.13}$ | $87.93_{\pm0.15}$ | $89.02_{\pm0.16}$ | $89.52_{\pm0.15}$ | $1725.16_{\pm11.24}$ | $55.47_{\pm2.72}$ | - |
| | VCD | $88.76_{\pm0.13}$ | $88.22_{\pm0.14}$ | $89.00_{\pm0.14}$ | $89.59_{\pm0.13}$ | $1704.11_{\pm11.06}$ | $58.27_{\pm2.43}$ | +1.00% |
| | VACoDe | $\mathbf{88.97}_{\pm0.08}$ | $\mathbf{88.47}_{\pm0.10}$ | $\mathbf{89.07}_{\pm0.09}$ | $\mathbf{89.63}_{\pm0.08}$ | $1722.62_{\pm4.16}$ | $59.87_{\pm1.66}$ | +2.07% |
| | *Self-Aug* | $88.79_{\pm0.13}$ | $88.20_{\pm0.14}$ | $88.68_{\pm0.09}$ | $89.25_{\pm0.09}$ | $\mathbf{1726.77}_{\pm3.63}$ | $\mathbf{60.50}_{\pm0.64}$ | **+2.25%** |

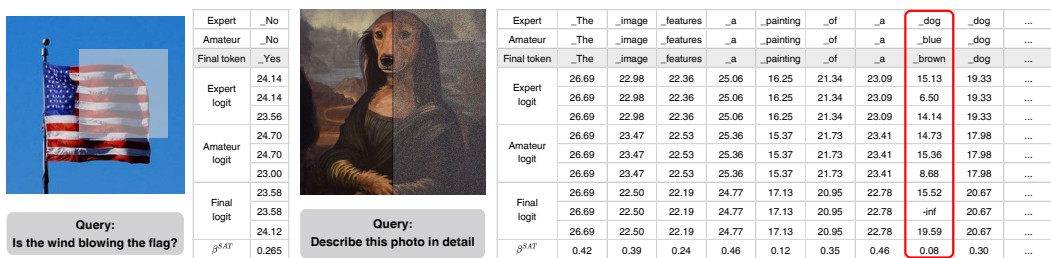

Figure 2: Qualitative examples of *Self-Aug* on MM-Vet (Yu et al., 2023) and LLaVA-Bench (Liu et al., 2023), and corresponding logit distributions and SAT thresholds by timestamp.

et al., 2014), and A-OKVQA (Schwenk et al., 2022) dataset, MME-Perception (MME-P) (Fu et al., 2024), and MMVP (Tong et al., 2024). MMHal-Bench (Sun et al., 2023), LLaVA-Bench (In-the-Wild) (Liu et al., 2023), MM-Vet (Yu et al., 2023) are selected for generative benchmarks. The ablation studies were focused on the LLaVA-1.5-7B the MME-P benchmark. This selection represents a methodological choice, as LLaVA-1.5 being one of the most widespread adoption within the open-source community, while MME-Perception offers the largest testbed among ones with diverse categories.

**Implementation Details** Unless explicitly stated otherwise, the CD hyperparameters are set to $\alpha = 1$, $\beta = 0.1$ for APC, and the SAT hyperparameter was set to $\gamma = -0.5$. Including automated judging of generative benchmarks, all API calls to proprietary models were made using OpenAI gpt-4o-mini with temperature 0 for deterministic results and reproducibility. All main experiments were conducted over five runs, and ablation studies over three runs with different random seeds, with results reported as the average and standard deviation to account for the inherent randomness from the augmentation process and multinomial sampling.

## 4.2 EXPERIMENTAL RESULTS

**Main Results** Tab. 1 and 2 summarize the averaged performance and standard deviations for all evaluated settings. The final column in each table, denoted as Avg. Δ, reports the average performance gain over the multinomial sampling baseline for each method and combination. For this calculation, the accuracy score is used for POPE and the average score is used for MMHal-Bench. For each configuration, the best-performing method is highlighted in bold, and ties are resolved in favor of

Table 2: Generative benchmark results on LLaVA-Bench (In-the-Wild) (Liu et al., 2023), MM-Vet (Yu et al., 2023), and MMHal-Bench Sun et al. (2023).

| Model | Method | MMHal-Bench Avg. Score$^\uparrow$ | MMHal-Bench Hal. Rate$_\downarrow$ | MM-Vet$^\uparrow$ | LLaVA-Bench$^\uparrow$ | Avg. $\Delta$ |
|-------|--------|-----------|-----------|---------|-------------|--------|
| LLaVA-1.5-7B | Multinomial | $2.27_{\pm 0.08}$ | $0.65_{\pm 0.02}$ | $27.74_{\pm 2.01}$ | $58.48_{\pm 2.17}$ | - |
|  | VCD | $2.32_{\pm 0.09}$ | $0.65_{\pm 0.02}$ | $31.14_{\pm 1.15}$ | $69.08_{\pm 2.07}$ | +2.82% |
|  | VACoDe | $2.32_{\pm 0.09}$ | $0.64_{\pm 0.03}$ | $29.88_{\pm 1.94}$ | $69.12_{\pm 2.48}$ | +6.14% |
|  | *Self-Aug* | $\mathbf{2.55}_{\pm 0.11}$ | $\mathbf{0.59}_{\pm 0.03}$ | $\mathbf{31.14}_{\pm 0.95}$ | $\mathbf{69.22}_{\pm 1.80}$ | **+6.97%** |
| LLaVA-1.5-13B | Multinomial | $2.35_{\pm 0.18}$ | $0.65_{\pm 0.05}$ | $31.20_{\pm 1.78}$ | $69.48_{\pm 2.78}$ | - |
|  | VCD | $2.37_{\pm 0.24}$ | $0.64_{\pm 0.07}$ | $35.00_{\pm 1.61}$ | $73.62_{\pm 1.40}$ | +1.11% |
|  | VACoDe | $2.52_{\pm 0.16}$ | $0.61_{\pm 0.03}$ | $34.18_{\pm 0.98}$ | $74.56_{\pm 1.62}$ | +4.78% |
|  | *Self-Aug* | $\mathbf{2.53}_{\pm 0.09}$ | $\mathbf{0.60}_{\pm 0.03}$ | $\mathbf{36.62}_{\pm 1.33}$ | $\mathbf{76.24}_{\pm 0.83}$ | **+6.04%** |
| Qwen-VL | Multinomial | $2.21_{\pm 0.12}$ | $\mathbf{0.50}_{\pm 0.02}$ | $31.70_{\pm 1.76}$ | $35.98_{\pm 1.56}$ | - |
|  | VCD | $2.17_{\pm 0.08}$ | $0.51_{\pm 0.03}$ | $34.04_{\pm 1.21}$ | $38.78_{\pm 0.58}$ | +9.21% |
|  | VACoDe | $\mathbf{2.21}_{\pm 0.13}$ | $0.50_{\pm 0.03}$ | $35.42_{\pm 1.36}$ | $39.18_{\pm 1.75}$ | +10.47% |
|  | *Self-Aug* | $2.15_{\pm 0.06}$ | $\mathbf{0.50}_{\pm 0.02}$ | $35.98_{\pm 1.00}$ | $\mathbf{39.84}_{\pm 0.73}$ | **+17.09%** |
| InstructBLIP | Multinomial | $1.89_{\pm 0.16}$ | $0.69_{\pm 0.05}$ | $23.06_{\pm 0.59}$ | $53.24_{\pm 2.01}$ | - |
|  | VCD | $1.99_{\pm 0.14}$ | $0.70_{\pm 0.04}$ | $28.18_{\pm 1.35}$ | $\mathbf{58.30}_{\pm 1.38}$ | +4.99% |
|  | VACoDe | $2.03_{\pm 0.07}$ | $0.68_{\pm 0.02}$ | $27.40_{\pm 0.51}$ | $56.82_{\pm 3.06}$ | +9.17% |
|  | *Self-Aug* | $\mathbf{2.16}_{\pm 0.13}$ | $\mathbf{0.64}_{\pm 0.05}$ | $\mathbf{31.14}_{\pm 0.95}$ | $56.98_{\pm 2.13}$ | **+17.08%** |
| Qwen3-VL-8B | Multinomial | $4.56_{\pm 0.11}$ | $0.31_{\pm 0.02}$ | $62.82_{\pm 1.06}$ | $118.50_{\pm 2.36}$ | - |
|  | VCD | $4.52_{\pm 0.16}$ | $0.32_{\pm 0.04}$ | $63.48_{\pm 1.36}$ | $119.76_{\pm 1.14}$ | +0.44% |
|  | VACoDe | $4.63_{\pm 0.14}$ | $0.29_{\pm 0.03}$ | $\mathbf{65.08}_{\pm 1.10}$ | $121.06_{\pm 1.56}$ | +2.45% |
|  | *Self-Aug* | $\mathbf{4.67}_{\pm 0.04}$ | $\mathbf{0.29}_{\pm 0.01}$ | $64.50_{\pm 1.23}$ | $\mathbf{121.88}_{\pm 1.39}$ | **+2.62%** |

the method exhibiting lower variance across runs. *Self-Aug* achieves remarkable performance gains across both benchmark categories, up to 18.78% relative to the multinomial sampling.

To further probe the effectiveness of *Self-Aug*, a token-level analysis was conducted to verify how the proposed method mitigates hallucinations by examining the output logits of LLaVA-1.5-7B. Fig. 2 illustrates two examples of logit values of LLaVA-1.5-7B with *Self-Aug* on MM-Vet (Yu et al., 2023) and LLaVA-Bench (Liu et al., 2023). Amateur and Expert logit indicate the selected token with and without augmentation, and the final token, highlighted with gray, is the token that corresponds to the argmax of the contrasted logit. Note that the applied augmentations are stylized for visual clarity.

These examples provide three important observations. (1) The example to the left shows a case of *failure correction* where the contrastive process between two logits successfully elevates the score for the correct ⌞Yes token, making it the final answer. (2) The example on the right evidences *hallucination penalty*, where random noise triggered hallucination of generating ⌞blue token from the amateur logit. It is penalized through subtraction, causing its final score to fall below the SAT threshold and be removed from the candidate set. (3) Adaptive nature of the SAT threshold $\beta^{\text{SAT}}$ is observed, with a higher threshold applied to common tokens (*e.g.*, articles, prepositions) and a lower threshold applied to informative, lower-confidence tokens (*e.g.*, painting, red-boxed token). These findings highlight a clear validation of both core components of *Self-Aug*, confirming that not only *contextually relevant* augmentation selection with model knowledge can effectively amplify the output divergence by invalidating the premise of the question, but also the efficacy of *confidence-aware* SAT.

**Computational Overhead** The computational cost of *Self-Aug* was evaluated by comparing throughput (token/s) and latency (ms/token) against other CD-based methods. The analysis used the LLaVA-Bench with the LLaVA-1.5 family on an NVIDIA A100 GPU. Detailed results are presented in Tab. 3. The superscript + on *Self-Aug* denotes the full prompt configuration that includes reasoning and ICL, while − denotes a lightweight configuration without both components. The results show that the primary computational bottleneck for both adaptive methods is the augmentation choice process. VACoDe is a *brute-force* that requires a separate

Table 3: Decoding throughput (token/s) and latency (ms/token). Scale and # tok indicate model parameters and the number of generated tokens for multimodal query, respectively.

| Decoding | Scale | # tok | tok/s$^\uparrow$ | ms/tok$_\downarrow$ | Score |
|----------|-------|-------|--------|---------|-------|
| VCD | 7B | 9914 | 18.50 | 54.06 | $69.08_{\pm 2.07}$ |
|  | 13B | 8785 | 14.01 | 71.38 | $73.62_{\pm 1.40}$ |
| VACoDe | 7B | 8418 | 16.97 | 58.93 | $69.12_{\pm 2.48}$ |
|  | 13B | 8568 | 13.03 | 76.76 | $74.56_{\pm 1.62}$ |
| *Self-Aug*$^-$ | 7B | 8346 | 17.39 | 57.50 | $69.20_{\pm 2.12}$ |
|  | 13B | 8793 | 11.37 | 87.92 | $73.82_{\pm 1.65}$ |
| *Self-Aug*$^+$ | 7B | 10163 | 15.08 | 66.32 | $69.22_{\pm 1.80}$ |
|  | 13B | 8805 | 11.33 | 88.26 | $76.24_{\pm 0.83}$ |

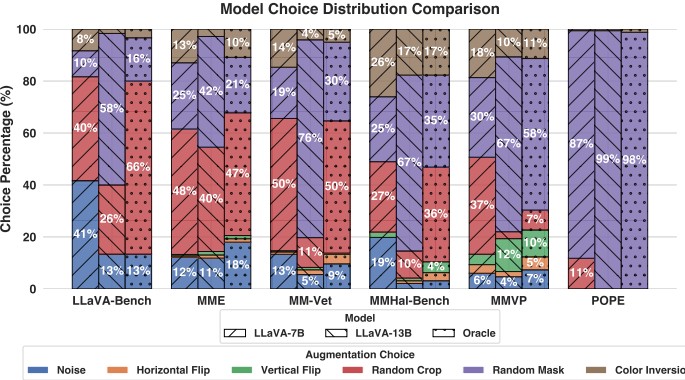

Figure 3: Distribution of self-augmentation choice across model size and benchmarks. Oracle indicates gpt-4o-mini decisions.

Table 4: Comparison between static augmentations and adaptive selection strategies on MME-P using LLaVA-1.5-7B.

| Augmentation | MME-P$^{\uparrow}$ |
|---|---|
| *Static* | |
| Noise | $1351.76_{\pm 7.59}$ |
| Horizontal flip | $1302.55_{\pm 47.62}$ |
| Vertical flip | $1354.42_{\pm 15.13}$ |
| Random crop | $1315.56_{\pm 41.75}$ |
| Random mask | $1302.50_{\pm 29.34}$ |
| *Adaptive* | |
| VACoDe | $1372.50_{\pm 13.78}$ |
| *Self-Aug* | $1431.30_{\pm 13.87}$ |
| Oracle | $1435.07_{\pm 22.30}$ |

forward pass for each predefined augmentation, which includes the full set of visual tokens, resulting in an overhead that scales linearly with the size of the augmentation set. On the other hand, *Self-Aug* demonstrates architectural advantage by requesting a *single text-only generation pass*, bypassing the main computational expenses of visual tokens, which constitute the majority of the input. This architectural feature enables a flexible trade-off between performance and latency, in that the cost-optimized prompting exhibits substantially higher efficiency with minimal impact on performance. This confirms that *Self-Aug* provides a more scalable and controllable framework for query-aware contrastive decoding.

### 4.3 ABLATION STUDY AND ANALYSES

**Augmentation Selection** A detailed investigation of the augmentation choice made by the LLaVA-1.5 family is presented in Fig. 3 with different patterns by model capacities. For comparison, the choices from gpt-4o-mini are also included to provide a practical upper bound and will be denoted as the "Oracle" for notation convenience throughout the remainder of this paper. The results reveal that the distribution of selections varies significantly across different benchmarks. A notable contrast is found between the sparsest POPE and the most uniform MMVP. For POPE, random mask accounts for 87.6% of all selections. This strong preference arises because the queries related to object recognition in POPE are unified as `Is there a {object} in the image?`, which are directly addressed by the definition of random mask as an occluding operation within the SAS Prompt. In contrast, the uniform distribution of MMVP reflects the diverse nature of the benchmark itself, which queries nine different *visual pattern* categories and thus applies a wider range of augmentations. On the other hand, the infrequent selection of horizontal flip across all benchmarks is a direct result of the evaluated queries rarely testing for horizontal spatial relationships. These findings suggest a broader principle that the set of predefined augmentations must be sufficiently diverse to match the complexity of the visual patterns in a given task, while noting that the specific distribution of choices is also dependent on the SAS Prompt design.

**Model Capacity** The impact of model scale on the quality of augmentation selection was evaluated by comparing the LLaVA-1.5 7B and 13B models against the Oracle baseline using two primary metrics. First, selection accuracy was measured by calculating the agreement with the Oracle choice, where the 7B model achieved 64.15% and the 13B model achieved 66.19%. Second, the quality of the reasoning trace for each choice was assessed by gpt-4o-mini on a scale of 0 to 10, with the 13B model producing higher quality justifications with an average score of 9.04 compared to 8.28 for the 7B model across benchmarks. The results from both metrics confirm that larger model capacity leads to improved augmentation selection and reasoning quality. Full prompt for reasoning assessment and detailed breakdown of these agreements are provided in the Appendix B.2 and E, respectively.

**Comparison with Single Augmentation** A comparison between static and adaptive visual augmentation strategies is presented in Tab. 4. Static strategies apply a single, fixed augmentation across all inputs, while adaptive strategies utilize query-aware augmentations. There is a clear performance gap between the two approaches, underscoring the importance of context-optimal augmentation. The

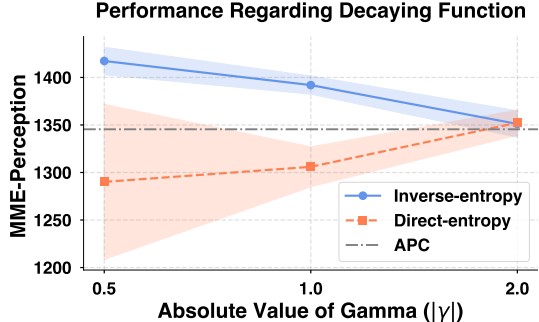

**Performance Regarding Decaying Function**

Figure 4: Comparison of the normalized entropy and proposed inverse-entropy function by $\gamma$.

Table 5: Plausibility constraint thresholding with APC and SAT.

| Decoding | Thresholding | MME-P$^\uparrow$ |
|---|---|---|
| VCD | APC ($\beta = 0.1$) | $1323.67_{\pm 20.84}$ |
| | SAT | $1395.17_{\pm 17.09}$ |
| VACoDe | APC ($\beta = 0.1$) | $1372.50_{\pm 13.78}$ |
| | SAT | $1414.21_{\pm 26.85}$ |
| *Self-Aug* | APC ($\beta = 0.1$) | $1345.46_{\pm 4.65}$ |
| | SAT | $1431.30_{\pm 13.87}$ |

significant gap between the proposed method and the others underscores the importance of *query-augmentation semantic alignment* and architectural flexibility, opening the possibility of leveraging diverse knowledge sources from internal knowledge to external reasoning modules.

**SAT Threshold** The core inverse-entropy heuristic of SAT was validated through a comparison of the proposed $H_{\text{decay}}$ against the APC baseline and a normalized scaled entropy $H_{\text{ns}}$ : $(-\sum_{i=0}^{|\mathcal{V}|-1}(p)_i \log_2(p)_i / \log_2 |\mathcal{V}|)^{1/\gamma}$. The $H_{\text{ns}}$ function is a direct-proportional entropy function, *i.e.*, implements the opposing rule to $H_{\text{decay}}$, mapping high input entropy to a more confined threshold. As visualized in Fig. 4, the results confirm a performance hierarchy: $H_{\text{decay}}$ consistently outperforms APC and $H_{\text{ns}}$ with more stable outputs. A further observation arises from the performance trend within each entropy-based function with respect to the scaling parameter $\gamma$. A lower $\gamma$ absolute value corresponds to a more restrictive threshold for $H_{\text{decay}}$ but a more generous one for $H_{\text{ns}}$. These findings not only imply mature thresholding is required to properly penalize false positives, but also provide strong empirical support for the *inverse-entropy principle* in the design of SAT.

Furthermore, the generalizability of SAT was evaluated through a direct comparison with the APC baseline. Both thresholding algorithms were applied to VCD, VACoDe, and *Self-Aug*, and the findings are presented in Tab. 5. The results show that SAT consistently outperforms APC across all decoding configurations, achieving an average performance gain of 4.94%. This performance gain is attributed to the foundational difference in the usage of model confidence. The consistency of this improvement suggests that SAT is broadly applicable to other CD-based methods.

**SAS Prompting** The individual contributions of the operational knowledge, reasoning and ICL components within SAS Prompting were evaluated by selectively ablating each component from the full prompt. As shown in Tab. 6, the provision of operational knowledge is the most critical component for performance, whereas reasoning and ICL have a marginal impact on accuracy. However, note that the reasoning instruction is the most primary factor affecting computational latency, as it requires the model to generate a full text sequence for the justification. Removing this instruction reduces the generation requirement to fewer than ten tokens for the final choice.

Table 6: SAS Prompting with and without reasoning steps and ICL. OK denotes the operational knowledge.

| OK | Reasoning | ICL | MME-P$^\uparrow$ |
|---|---|---|---|
| | Baseline | | $1278.42_{\pm 30.30}$ |
| ✗ | ✗ | ✗ | $1312.39_{\pm 11.71}$ |
| ✓ | ✗ | ✗ | $1419.08_{\pm 11.39}$ |
| ✓ | ✓ | ✗ | $1428.02_{\pm 7.92}$ |
| ✓ | ✗ | ✓ | $1428.63_{\pm 31.85}$ |
| ✓ | ✓ | ✓ | $1431.30_{\pm 13.87}$ |

## 5 DISCUSSION

**Failure Cases** *Self-Aug* inherits the fundamental assumptions of CD. Suppose that we have a false positive index $i$ in both expert logit $l$ and amateur logit $l'$. It is assumed that $l'$ has a higher likelihood on $i$-th index ($l[i] < l'[i]$) due to the disruption from visual augmentations. Then, the $i$-th value of the contrasted logit $l_{\text{CD}}$ will decrease since $(1+\alpha) \cdot l[i] - \alpha \cdot l'[i] < l[i]$ always

Table 7: Multinomial sampling results on Qwen3-VL-32B. High scores on $l'$ induce failures.

| Benchmark | $l$ | $l'$ | $l_{\text{CD}}$ |
|---|---|---|---|
| MMVP | $67.56_{\pm 1.39}$ | $68.44_{\pm 1.02}$ | $67.78_{\pm 0.38}$ |
| MME-P | $1798.15_{\pm 8.64}$ | $1786.43_{\pm 2.81}$ | $1789.94_{\pm 0.73}$ |

holds by the assumption $l[i] < l'[i]$. Conversely, for a true positive token index $j$, the method assumes the a lower likelihood on amateur logit ($l[j] > l'[j]$). If $l'$ remains as accurate as $l$, the contrastive formulation yields minimal performance gains because the two distributions are nearly identical. Therefore, the failure case in all CD-based methods would be the case when the two assumptions are invalidated (*i.e.*, $l[i] > l'[i]$ and $l[j] \leq l'[j]$). Tab. 7 demonstrates this phenomenon using the averaged scores of Qwen3-VL-32B across three separate runs. Determining and addressing such failure is beyond the scope of this study, and is deferred to future works.

**Limitations and Future Work** The proposed method presents several branches for future work by addressing current limitations. First, the effectiveness of SAS Prompting depends on the reasoning and instruction-following ability of the base model. Less capable models might produce malformed outputs or poor augmentation choices. This dependency could be addressed in future work by developing more robust prompting methods (*e.g.*, Chain-of-Thoughts (Wei et al., 2022)) or utilizing a smaller, specialized model for the selection task. Second, the current method is limited to a predefined set of visual augmentations. While this set covers common scenarios, it may not contain the best augmentation for highly specialized visual reasoning tasks. A promising solution involves developing methods that can dynamically select from a more diverse and larger library of transformations using external modules (*e.g.*, object detector), enhancing the versatility. Finally, expanding the principle to the video domain by designing temporal-aware dynamic recalibration methods of obtaining contrastive pairs would enable frame-consistent decoding. The direction aids understanding complex temporal contexts and facilitating downstream applications as highlighted in previous studies (Im et al., 2023; Ali et al., 2024).

## 6 CONCLUSION

This study introduces *Self-Aug*, a novel decoding strategy designed to mitigate hallucinations in LVLMs. The proposed method aligns the semantics between query and visual augmentation by leveraging the flexible intrinsic reasoning of the model without relying on predefined heuristics. In addition, the proposed sparsity adaptive truncation introduces a confidence-aware thresholding that dynamically adjusts candidate sets based on logit entropy, effectively penalizing false positives. Extensive experiments conducted across five LVLM and seven benchmarks demonstrated that *Self-Aug* consistently improves factual consistency over existing decoding strategies while maintaining practical computational efficiency. Beyond immediate performance gains, this study underlines the importance of the semantic coupling of query-augmentation with confidence-sensitive decoding as a principled approach for developing more robust multimodal generation.

### ACKNOWLEDGMENTS

We gratefully acknowledge the Complex Data Reasoning and Analysis Lab at Arizona State University for their resources and computational support.

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

# *Self-Aug*: QUERY AND ENTROPY ADAPTIVE DECODING FOR LARGE VISION-LANGUAGE MODELS

## APPENDIX

Due to space limitations in the main manuscript, we provide supplementary materials in this appendix that elaborate on the proposed design, experimental settings, and visualizations. This includes the complete prompt design for Self-Augmentation Selection (SAS) and LLM-as-a-Judge for reasoning quality, additional qualitative examples, extended experimental results, and a detailed breakdown of the model and benchmark information.

## A    ADDITIONAL EXPERIMENTAL SETUP DETAILS

The visual augmentations were implemented on top of the official VCD (Leng et al., 2024) source code with the following specific parameters. The color inversion operation was performed using the PyTorch `torchvision.transforms.functional.invert` function. For the random crop and random mask augmentations, a ratio of 2.0 was used, which corresponds to applying the operation to a randomly placed square patch with side lengths equal to half the original image dimensions. For the noise augmentation, a diffusion noise step of 500 was applied from the official VCD random noise implementation.

## B    FULL PROMPT DESIGN

This section provides the verbatim prompts used for both the self-augmentation selection and the subsequent reasoning quality assessment. The full SAS Prompt, which leverages in-context learning and reasoning to achieve optimal query-augmentation semantic alignment, is presented first. This is followed by the prompt used to instruct the LLM-as-a-Judge for the evaluation of the SAS reasoning trace against the Oracle. The individual effects of the reasoning and in-context learning components within the SAS Prompt are quantified in the ablation study section of the main manuscript.

### B.1    SAS PROMPTING

You are an expert data augmentation analyst. Your task is to select the single most semantically disruptive image augmentation that most effectively invalidates the question's premise or prevents a confident answer. Provide a clear reason explaining why the augmentation is chosen, then state your final choice.

## Augmentations and Their Effects ##
- Vertical flip: Flips image top-to-bottom. Disrupts questions about "above", "below", "under" or reading orientation.
- Color inversion: Replaces each color with its complement. Disrupts questions relying on accurate color identification.
- Random crop: Removes random parts of the image. Disrupts questions requiring global context or peripheral objects.
- Random mask: Occludes portions of the image. Disrupts object presence, count, or attribute recognition.
- Noise: Adds visual distortion. Disrupts questions requiring small details, texture, or text clarity.
- Horizontal flip: Flips the image left-to-right. Disrupts questions about left/right positioning and left-to-right text reading.

## Examples ##
Question: "Is the mirror above the TV?" Reason: The question focuses on vertical positioning. Vertical flip reverses top and bottom, making "above" mean "below," invalidating the question. Other augmentations don't affect vertical relationships. Choice: vertical flip

Question: "Is this photo taken indoors?" Reason: The question requires identifying a specific environmental context. Random crop may exclude key background elements like trees, invalidating the question. Flips, color inversion, noise, and random mask don't directly affect scene context. Choice: random crop

Question: "Are there any green beans in the image?" Reason: The question requires identifying a specific color. Color inversion changes green to its complement, invalidating the question. Flips, noise, random mask, and random crop don't target color directly. Choice: color inversion

Question: "How many people are in the image?" Reason: The question requires counting visible people. Random mask can completely obscure one or more people, making the exact count impossible. Noise obscures details but typically doesn't hide entire objects, allowing approximate counting. Flips and color inversion don't affect object visibility or count. Choice: random mask

Question: "Is the cat on the right side of the laptop?" Reason: The question relies on horizontal positioning. Horizontal flip reverses left and right, making "right" mean "left", invalidating the question. Other augmentations don't target horizontal positions. Choice: horizontal flip

Question: "Does this artwork exist in the form of painting?" Reason: The question requires identifying the texture of the artwork. Noise obscures fine details, making it hard to identify the medium. Other augmentations don't target texture details. Choice: noise

## Your Answer ##
If multiple augmentations could disrupt the question, select the one whose effect is most direct and unambiguous. You must choose one of the given augmentations following the "Reason:" and "Choice:" format.
Question: "{text}"

## B.2    LLM-AS-A-JUDGE PROMPT FOR REASONING QUALITY

Your task is to evaluate a candidate model's response against an expert-provided reference solution. The goal is to select the image augmentation that most effectively disrupts the premise of a given question.

## Evaluation Rubric (Integer Scale 0-10) ##
- 10 (Excellent): The candidate's choice is highly effective and the reasoning is clear, logically sound, and directly supports the choice. The response is of reference quality.
- 7-9 (Good): The choice is effective and the reasoning is logical, but may be slightly less specific or insightful than the reference.
- 4-6 (Acceptable): The choice is plausible but not optimal. The reasoning is generic, weak, or contains minor flaws.
- 1-3 (Poor): The choice is ineffective and the reasoning is flawed or irrelevant.
- 0 (Very Poor): The choice and reasoning are completely incorrect or nonsensical.

## Reference Example ##
Question: "How many people are in the image?"
Reference Reason: "The question requires counting visible people. Random mask can completely obscure one or more people, making the exact count impossible."
Reference Choice: "random_mask"
Candidate Reason: "Random crop might cut some people out of the frame."
Candidate Choice: "random_crop"
Evaluation: Score: 7, Reason: The candidate's choice is a valid strategy for disrupting a counting task, but it is less direct than the reference. The reasoning is correct but lacks specificity.

## Task ##
Question: "{question}"
Reference Reason: "{oracle_reason}"
Reference Choice: "{oracle_choice}"
Candidate Reason: "{model_reason}"
Candidate Choice: "{model_choice}"

Evaluation:

## C    MODEL AND BENCHMARK DETAILS

**Model Families**

- **LLaVA-1.5** (Liu et al., 2023) is a powerful open-source LVLM that establishes the effectiveness of visual instruct tuning for creating general-purpose visual assistants. Its architecture is characterized by its simplicity, connecting a pretrained CLIP vision encoder to a Vicuna LLM using a single Multi-Layer Perceptron projection layer. The LLaVA-1.5 version improved upon the original by incorporating a more capable LLM and scaling the instruction-following data.

- **Qwen-VL** (Bai et al., 2023) is a series of highly performant, versatile vision-language models based on the Qwen language model family. A key feature of the Qwen-VL architecture is its support for multiple languages, the ability to process multi-image inputs, and its strong capabilities in fine-grained visual understanding, including text recognition and object localization.

- **InstructBLIP** (Dai et al., 2023) is a vision-language instruction tuning framework designed to enhance zero-shot generalization across a diverse set of tasks. Its central innovation is the use of an instruction-aware Query Transformer. This module is trained to extract visual features from the image encoder that are specifically relevant to the given text instruction, enabling more targeted and effective multimodal reasoning.

- **Qwen3-VL** (Bai et al., 2025) is the latest evolution in the Qwen vision-language series, engineered to overcome the limitations of fixed-resolution processing and enhance visual agent capabilities. Its architecture employs Naive Dynamic Resolution, which processes images at their native aspect ratios by converting them into variable-length token sequences, effectively eliminating the loss of detail caused by resizing or padding. Combined with Multimodal Rotary Positional Embeddings (M-RoPE) that decompose positional information into temporal, height, and width components, the model achieves state-of-the-art performance in comprehending high-resolution documents, long-context videos, and executing complex tool-use instructions.

**Discriminative Benchmarks**

- **MME** (Fu et al., 2024) is a benchmark that provides a granular evaluation of multimodal tasks, spanning 10 perception and 4 cognition categories. The performance is measured on binary yes or no questions using an accuracy-based MME score. Following the standard practice (Leng et al., 2024; Kim et al., 2024b), we consider the perception category for the experiments.

- **MMVP** (Tong et al., 2024) is designed to evaluate a model's understanding of fine-grained visual details. It achieves this by using 300 CLIP-blind image pairs, where models must capture subtle differences to perform paired classification accurately. These image pairs cover nine distinct visual patterns: orientation and direction, feature presence, state and condition, quantity and count, positional and relational context, color and appearance, structural and physical characteristics, text, and viewpoint and perspective. The evaluation follows a multiple-choice format, where final model responses are mapped to the answer options using GPT-4 as an automated judge.

- **POPE** (Li et al., 2023d) serves as a dominant benchmark for assessing object hallucination by testing models with three distinct types of negative questions. These categories include queries about random non-existent objects, popular objects that are frequent in the dataset but absent from the image, and adversarial objects selected for their high co-occurrence. The dataset contains 9,000 question-image pairs built from 500 images, each evaluated against multiple questions across the three categories.

**Generative Benchmarks**

- **LLaVA-Bench (In-the-Wild)** (Liu et al., 2023) is a benchmark to evaluate the ability of Large Vision Language Models (LVLMs) to handle complex tasks and adapt to new domains. It features 24 images and 60 queries, which collapse into three categories: conversation,

detailed description, and complex reasoning. The evaluation is conducted using GPT-4V as a judge to rate both the model response and a reference answer. The final performance is reported as a score ratio, calculated by dividing the total score of the reference answer.

- **MMHal-Bench** (Sun et al., 2023) evaluates and penalizes hallucinations across a diverse set of reasoning types. It is composed of 96 image-question pairs that cover eight distinct categories, including object attributes, comparison, and spatial relations. Evaluation is performed using GPT-4V as an automated judge to assess the severity of hallucination in the generated response. The responses are scored on a scale from 0 to 7, where a higher score indicates greater facutal consistency.

- **MM-Vet** (Yu et al., 2023) evaluates an LVLM to integrate multiple multimodal capabilities for complex reasoning. The benchmark defines six fundamental multimodal abilities: recognition, knowledge, OCR, spatial awareness, language generation, and mathematics. A key feature of MM-Vet is its focus on compositional tasks, where these six core abilities are combined to create 16 distinct capability integrations. The dataset itself is composed of 200 images and 218 questions, each requiring a specific combination of these integrated skills.

## D    ADDITIONAL QUALITATIVE RESULTS

To provide a more granular understanding of the behavior of the method, this section presents additional qualitative results from both discriminative and generative benchmarks. Each example provides a comprehensive analysis that includes the reasoning trace for the chosen augmentation, a stylized visualization of the augmentation, the logit values for the expert, amateur, and contrasted distributions, and the corresponding Sparsity Adaptive Truncation (SAT) threshold. For improved visualization clarity, common punctuation tokens such as commas and periods have been omitted from the presented logit distributions.

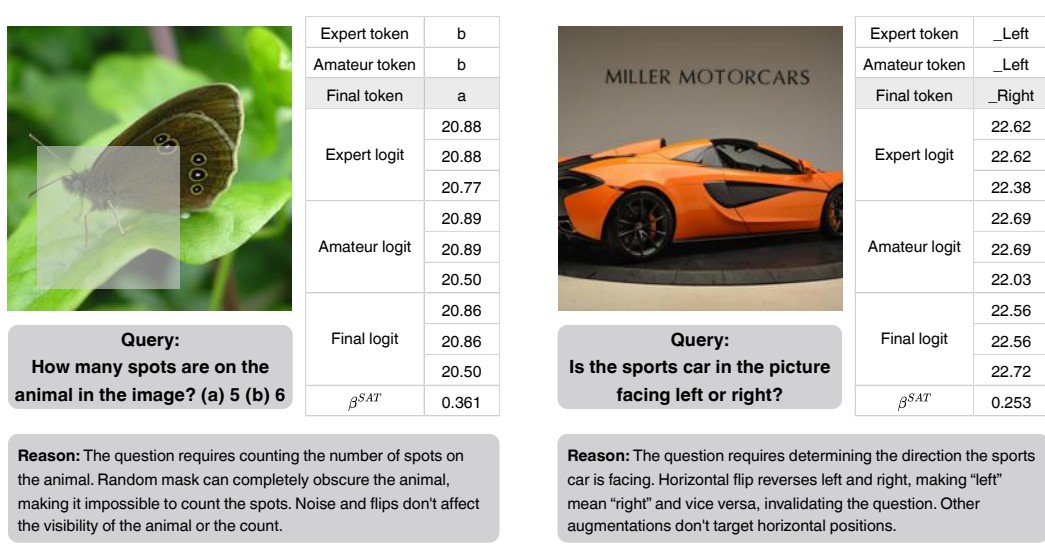

Figure A.1: Qualitative results on MMVP (Tong et al., 2024).

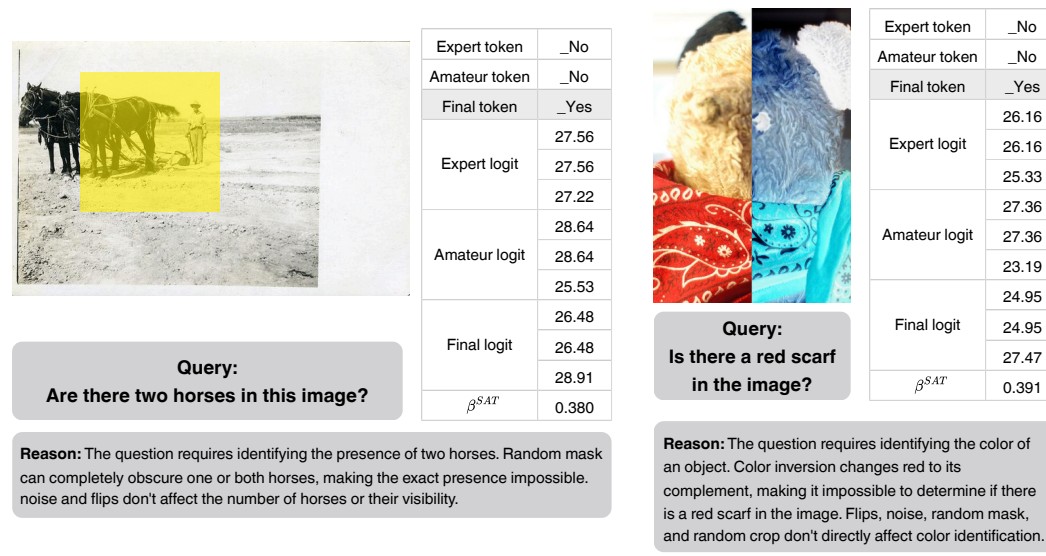

Figure A.2: Qualitative results on MME (Fu et al., 2024).

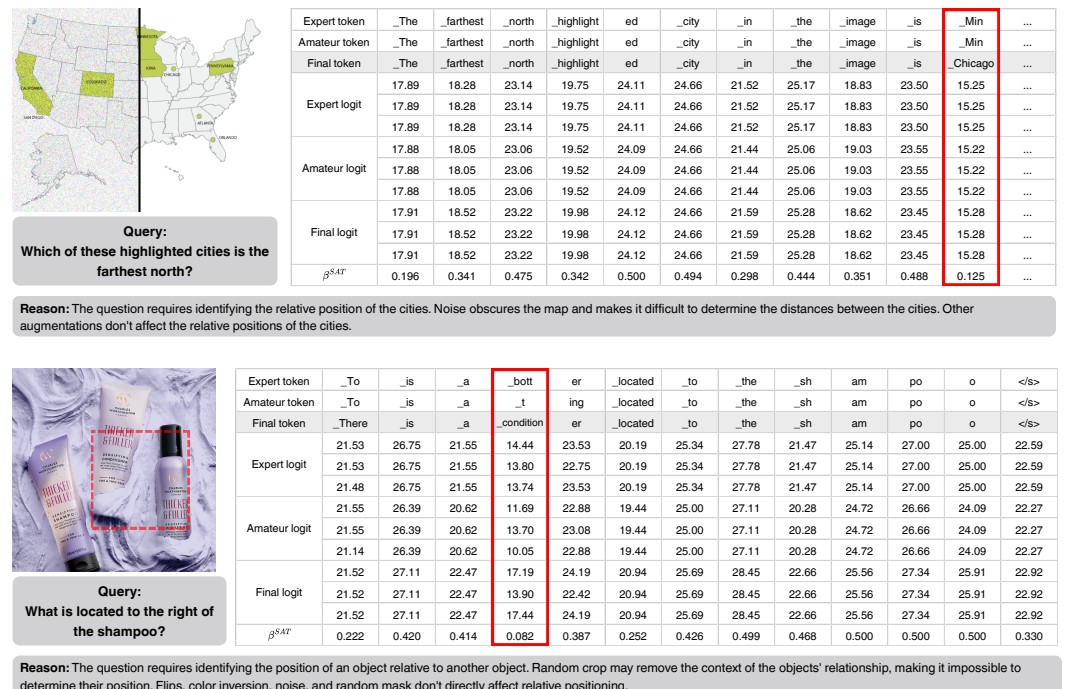

Figure A.3: Qualitative results on MM-Vet (Yu et al., 2023).

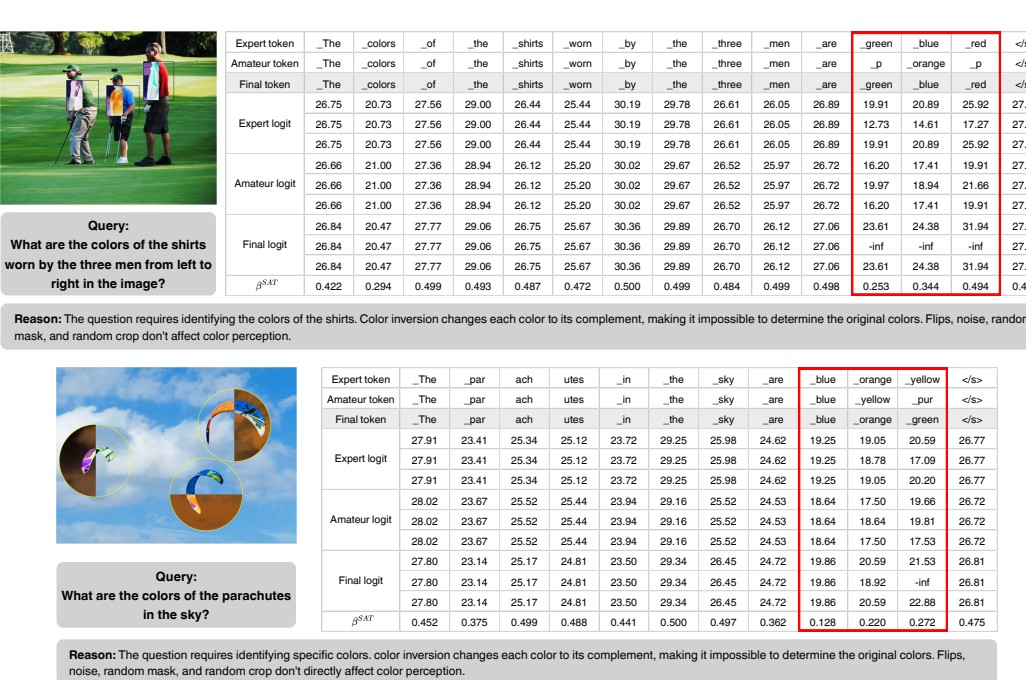

Figure A.4: Qualitative results on MMHal-Bench (Sun et al., 2023).

Figure A.5: Confusion matrix for LLAVA-1.5 7B model.

Figure A.6: Confusion matrix for LLAVA-1.5 13B model.

Table A.1: Comparison of model performance against the gpt-4o-mini oracle. Agreement measures the accuracy percentage (%) of augmentation choices, and Judge Score estimates the quality rating of the model reasoning on a 0 to 10 scale by gpt-4o-mini.

| Model | Metric | LLaVA-Bench | MME | MM-Vet | MMHal | MMVP | POPE | Average |
|---|---|---|---|---|---|---|---|---|
| LLaVA-7B | Agreement (%) | 56.67 | 72.56 | 61.93 | 54.17 | 52.00 | 87.57 | 64.15 |
| | Judge Score | 7.97 | 8.59 | 7.98 | 7.79 | 7.69 | 9.63 | 8.28 |
| LLaVA-13B | Agreement (%) | 51.67 | 69.77 | 40.37 | 63.54 | 72.67 | 99.10 | **66.19** |
| | Judge Score | 8.63 | 9.12 | 8.55 | 9.08 | 8.86 | 9.99 | **9.04** |

# E    DETAILED COMPARISON AGAINST ORACLE

In the main script, experiments were conducted to evaluate the impact of the model scale on the quality of augmentation choice and reasoning. The agreement of each model's choice and the quality of its reasoning trace were measured against the Oracle, with the results summarized in Tab. A.1. These results confirm that larger model capacity generally leads to better query-augmentation semantic alignment and higher reasoning quality.

A more granular analysis using the confusion matrices in Fig. A.5-A.6, reveals a complex, task-dependent relationship. On uniform benchmarks such as POPE, the alignment between the 13B model and the Oracle is nearly optimal. In contrast, on more complex benchmarks such as MM-Vet, the 13B model exhibits a predictive bias, frequently selecting random crop when the Oracle chooses the functionally similar random mask. Note that this disagreement is not a critical failure, but rather a choice between two functionally similar occlusion-based augmentations.

This finding highlights a key strength of the proposed method. The fact that strong downstream performance is achieved without requiring a perfect, Oracle-level selection confirms that the framework is highly effective at leveraging the competent, albeit imperfect, reasoning of different model scales to significantly improve factual consistency.

