# OpenReview forum: "Self-Aug: Query and Entropy Adaptive Decoding for Large Vision-Language Models"
_ICLR.cc/2026/Conference — ICLR 2026 Poster_

### Official Review · Reviewer_poDY · 2025-10-29

**Soundness:** 3
**Presentation:** 3
**Contribution:** 2
**Rating:** 6
**Confidence:** 3

**Summary:**

This paper tackles hallucination in Large Vision-Language Models (LVLMs) by improving visual contrastive decoding (VCD). It introduces SAVCD: (1)a self-augmentation selection (SAS) prompt that asks the LVLM to pick a semantically-disruptive augmentation for the current query, and (2) Sparsity Adaptive Truncation (SAT) — a thresholding rule that uses the entropy of the expert logit distribution to set the plausibility cutoff. At each step, SAVCD contrasts the expert and amateur logits, then removes tokens below the SAT threshold before sampling. Experiments on multiple LVLM families and 7 benchmarks show consistent gains over prior VCD methods.

**Strengths:**

1. The technical soundness of SAS and SAT is solid. Authors have conducted quantitative experiments and intuitive explanations to demonstrate its effectiveness.
2. The experiments, though tested on old models, can be rated as solid.
3. The paper is well-written: clear and easy to follow. The figures are carefully polished and informative.

**Weaknesses:**

1. The contributions sound incremental, given that VCD has already been widely researched. SAS and SAT make sense intuitively and experimentally, but might not be enough contribution for a top tier conference. However, I am not an expert on LLM Decoding, so I will carefully refer to other reviewers' comments on novelty and contribution before making my final decision.
2. The SAS Augmentation Choice seems arbitrary to me, as "Noise, Horizontal Flip, Vertical Flip, Random Crop, Random Mask and Color Inversion" only presents a subset of the image's properties. For instance, when asked about size, reachability, OCR or spatial relationship, chances are that these Augmentation choices will not work.
3. The experimental results don't show robust generalization across datasets and models: for instance, SAVCD in Qwen-VL or POPE-MSCOCO consistently underperform other methods.
4. The experiments are conducted on rather old models like LLaVA-1.5, Qwen-VL, which may cast doubt on its effectiveness on its effect on contemporary models. The claim could be strengthened with an experiment on Qwen2.5-VL.

(minor) For Table 1: LLaVA-1.5-7B (POPE-MSCOCO) Acc. SOTA should be VACoDe, not SAVCD.

**Questions:**

I would appreciate it if the authors can share any information about applying it to newer models, if they had done experiments on that before. This would help me determine if this method has value for today's VLM development.

I am giving a slightly higher score, as I hope authors can address my concerns.

---

> ### Author Response · Authors · 2025-11-20
>
> We truly appreciate your comments and finding our paper to be technically solid, includes extensive experiments, and well-written. We would like to address your concerns point by point below.
>
> ---
>
> **Q1.** The contributions sound incremental, given that VCD has already been widely researched. SAS and SAT make sense intuitively and experimentally, but might not be enough contribution for a top tier conference. However, I am not an expert on LLM Decoding, so I will carefully refer to other reviewers' comments on novelty and contribution before making my final decision.
>
> **A1.** Thank you for your comment.
> We would like to clarify the distinct novelty of our two key contributions.
>
> First, VCD [1] and following works [2,3] focus on finding the contrastive image counterpart by modifying the input image in **identical underlying heuristics**.
> Even though the counterpart varies by factors such as randomness (e.g., random noise), the main functionality of the algorithms remains **static** and query-agnostic, failing to adapt to the different cognitive tasks required by diverse queries.
> While VACoDe [4] also proposed a dynamic visual augmentation strategy, new limitations on its brute-force mechanism and dependence on the first token emerged as mentioned in lines 55 and 132.
> Also, they use different augmentation sets by benchmark in practice.
> SAS Prompt is proposed to address the limitations of dynamic VCD in the current state by **coupling the query intent and visual derivatives**, with more controllable and flexible computational overhead.
>
> Separately, this work improves APC to SAT.
> Contrastive decoding may undesirably reward false negatives; APC filters them by setting a threshold, which is determined by subtracting a **fixed** parameter from the max logit value.
> However, static thresholding may result in improper pruning depending on the overall sparsity of the logit since **each logit value is relative to each other in the distribution**.
> SAT mitigates this limitation by leveraging entropy, a relational information between the logit values within the distribution.
> This enables SAT to calculate a dynamic threshold that correctly adapts its leniency based on the model's objective state of uncertainty, avoiding the improper pruning that plagues static, confidence-agnostic filtering of APC.
>
> We will further clarify these in the camera-ready version of our manuscript to highlight novelty.
>
> [1] Leng et al. Mitigating object hallucinations in large vision-language models through visual contrastive decoding. CVPR 2024
>
> [2] Huang et al. Opera: Alleviating hallucination in multi-modal large language models
> via over-trust penalty and retrospection-allocation. CVPR 2024
>
> [3] Wang et al. Mitigating hallucinations in large vision-language models with instruction contrastive decoding. ACL 2024
>
> [4] Kim et al. Vacode: Visual augmented contrastive decoding. arXiv preprint arXiv:2408.05337

---

> ### Author Response · Authors · 2025-11-20
>
> **Q2.** The SAS Augmentation Choice seems arbitrary to me, as "Noise, Horizontal Flip, Vertical Flip, Random Crop, Random Mask and Color Inversion" only presents a subset of the image's properties. For instance, when asked about size, reachability, OCR or spatial relationship, chances are that these Augmentation choices will not work.
>
> **A2.** Thank you for your comment.
> We acknowledge that the predefined set of six augmentations is a limitation, as mentioned in the "Limitations and Future Work" section.
> This set was adopted from the established methodology in prior work [1], which has already shown that these specific transformations are effective at targeting a diverse range of visual capabilities.
> This set is sufficiently diverse because random noise hinders holistic and compositional understanding, both flips can invalidate any spatial-related queries, color and appearance disrupt attributal perception, and random masking and cropping can be applied broadly since they occlude the object-of-interest and background information, respectively.
> We would like to further demonstrate how the augmentation set has wide coverage and is sufficiently diverse using nine different visual patterns identified in the MMVP [2] paper as an example.
>
> - Color and Appearance: Challenged by color inversion, which forces the model to identify objects by attributes and context rather than relying on learned color properties.
>
> - Orient\&Direction, Positional\&Relational Context, Viewpoint\&Perspective: Directly invalidated by horizontal and vertical flip.
>
> - Text (OCR), State/Condition, and Structural Characteristics: Random noise impedes holistic understanding of the scene.
>
> - Presence of Specific Features and Quantity\&Count: Hindered by occlusion-based operations (random mask and crop).
>
> These coverages demonstrate that, despite its compact size, the augmentation pool covers a wide spectrum of visual stressors relevant to multimodal understanding. Extending this pool beyond a predefined set remains an important direction for future work, and we intend to explore more adaptive and data-driven augmentation mechanisms as part of our subsequent research.
>
> [1] Kim et al. Vacode: Visual augmented contrastive decoding. arXiv preprint arXiv:2408.05337
>
> [2] Tong et al. Eyes wide shut? exploring the visual shortcomings of multimodal llms. CVPR 2024
>
> ---
>
> **Q3.** The experimental results don't show robust generalization across datasets and models: for instance, SAVCD in Qwen-VL or POPE-MSCOCO consistently underperform other methods.
>
> **A3.** Thank you for your comment.
> We would like to reinforce the empirical claim that SAVCD demonstrates robust generalization and significantly outperforms other VCD methods across diverse architectures.
> The averaged performance gain of our method demonstrates its superior ability to handle varying cognitive tasks:
>
> | Decoding | Avg. Gain |
> | :--- | --- |
> | VCD | 7.43% |
> | VACoDe | 9.20% |
> | SAVCD | 12.64% |
>
> **Table 1.** Performance gain on Qwen-VL across all tested benchmarks.
>
> ---
>
> **Q4.** The experiments are conducted on rather old models like LLaVA-1.5, Qwen-VL, which may cast doubt on its effectiveness on its effect on contemporary models. The claim could be strengthened with an experiment on Qwen2.5-VL.
>
> **A4.** Thank you for your suggestion.
> We anticipate that most of the LVLMs can benefit from our decoding method since they share similar underlying structures, pre/post-training mechanisms across the model.
> To further empirically verify this claim and demonstrate the broad applicability of our method, we conducted additional experiments on the Qwen3-VL-8B and 32B, which represent a state-of-the-art LVLM family.
> Due to time and resource constraints, we report results 5 runs on MMVP, and LLaVA-Bench.
> SAVCD consistently outperforms the other decoding methods as shown below.
> We will add the full experimental results on the new version of our manuscript.
>
> | Model | Decoding | MMVP | LLaVA-Bench |
> | :--- | :--- | :---: | :---: |
> | Qwen3-VL-8B | Multinomial | $55.87_{\pm 1.19}$ | $120.42_{\pm 1.90}$ |
> | Qwen3-VL-8B | VCD | $56.27_{\pm 1.80}$ | $120.40_{\pm 1.36}$ |
> | Qwen3-VL-8B | VACoDe | $55.20_{\pm 0.56}$ | $120.60_{\pm 0.99}$ |
> | Qwen3-VL-8B | SAVCD | $57.87_{\pm 1.59}$ | $121.84_{\pm 0.75}$ |
> | Qwen3-VL-32B | Multinomial | $53.73_{\pm 1.19}$ | $122.66_{\pm 1.88}$ |
> | Qwen3-VL-32B | VCD | $55.87_{\pm 0.56}$ | $122.54_{\pm 1.13}$ |
> | Qwen3-VL-32B | VACoDe | $52.00_{\pm 1.94}$ | $121.34_{\pm 1.74}$ |
> | Qwen3-VL-32B | SAVCD | $57.20_{\pm 2.13}$ | $123.18_{\pm 1.17}$ |
>
> **Table 2.** Results with Qwen3-VL on MMVP and LLaVA-Bench.
>
> ---
>
> **Q5.** For Table 1: LLaVA-1.5-7B (POPE-MSCOCO) Acc. SOTA should be VACoDe, not SAVCD.
>
> **A5.** Thank you for pointing out the error. We will update them with the full experimental results on Qwen3-VL.

---

> > ### Comment · Reviewer_poDY · 2025-11-25
> > **Experiments convincing; remain supportive scoring**
> >
> > As the authors have clearly demonstrated their superiority with new experiments, my concerns about the generalization of SAVCD are addressed. I find the experiments on Qwen3-VL convincing, and I believe putting that table into main paper will be beneficial if the paper is accepted.
> >
> > The two novelties explained are not fundamentally innovative or impactful; however, given the strong performance, I keep my scoring of 6.

---

> > > ### Author Response · Authors · 2025-11-26
> > >
> > > We sincerely thank the reviewer for their thoughtful engagement throughout the review process. We greatly appreciate the constructive feedback and insightful comments regarding a deeper investigation into the underlying mechanisms and potential solutions. We will bear these in mind when revising our paper for future publication. Thank you once again for your time and valuable guidance.

---

### Official Review · Reviewer_msRD · 2025-10-31

**Soundness:** 3
**Presentation:** 3
**Contribution:** 2
**Rating:** 4
**Confidence:** 4

**Summary:**

This paper proposes a training-free decoding strategy named SAVCD, aimed at mitigating hallucinations in Large Vision-Language Models (LVLMs). The method features two core contributions: 1) a Self-Augmentation Selection (SAS) prompting strategy that leverages the model's own knowledge to dynamically select the most relevant visual augmentation for constructing a meaningful "amateur" model in contrastive decoding; and 2) a Sparsity Adaptive Truncation (SAT) algorithm that dynamically adjusts the threshold for candidate tokens based on the entropy of the output logit distribution to improve factual consistency. Experimental results demonstrate that the proposed method outperforms existing decoding strategies across several benchmarks and models.

**Strengths:**

1.  The paper presents a complete and well-designed decoding framework. The experimental section is thorough, covering multiple models and benchmarks, and the results consistently show that the proposed SAVCD method outperforms baselines in improving factuality. This demonstrates its practical utility as a plug-and-play tool.
2.  The core idea of having the model self-select the most appropriate visual augmentation via prompt engineering to construct an "amateur" perspective for contrastive decoding is highly novel and interesting. It explores the possibility of using a model's own reasoning capabilities to guide its decoding process, offering a new perspective on mitigating LVLM hallucinations.
3.  As a purely inference-time strategy, SAVCD requires no additional training or architectural modifications. This gives it strong generality and a low barrier to adoption, making it an attractive feature for the community.

**Weaknesses:**

The main weakness of this paper lies in its methodology, which is built on an assumption that may not always hold, and the solution relies to some extent on elaborate engineering tricks, which may affect its robustness and generality.

1.  **Questionable Core Methodological Assumption**: The core of SAVCD, the SAS prompt, relies on a critical assumption: that a model prone to hallucination and reasoning failures can accurately interpret a complex prompt to select an optimal corrective strategy for itself. This introduces a degree of circular reasoning. If the model's foundational reasoning is strong enough to perform this selection task perfectly, the initial problem of hallucination might be less severe. This design makes the method's success partially dependent on the very problem it aims to solve.
2.  **Over-reliance on Prompt Engineering**: The method's effectiveness is heavily dependent on a meticulously crafted, complex prompt that includes extensive descriptions and examples. Such approaches are often brittle, being sensitive to changes in wording, structure, or even model versions. Furthermore, its scalability is limited; for instance, introducing new augmentation types would require manually modifying and debugging this complex prompt. This makes the method feel more like a highly-tuned heuristic for a specific set of tasks and models rather than a general, robust principle.

**Questions:**

1.  Regarding the robustness of SAS selection: What is the accuracy of the model's augmentation selection in the first step? Are there cases where the model selects a suboptimal or incorrect augmentation, leading to worse results than the baseline? Could the authors provide an analysis of such failure cases?
2.  Regarding the limitations of the SAT algorithm: The core of SAT is using entropy to measure model uncertainty. However, models can sometimes produce a completely wrong hallucination with very high confidence (i.e., very low entropy). In such "confidently wrong" scenarios, could SAT's stricter threshold inadvertently lock in this error, making it harder to correct?
3.  Regarding the generality of the prompt: The SAS prompt appears quite elaborate. How much would the performance degrade if this prompt were simplified (e.g., providing only the names and simple descriptions of augmentations, without the ICL examples)? This would help in understanding to what extent the performance gain comes from the method itself versus the elaborate prompt engineering.

---

> ### Author Response · Authors · 2025-11-20
>
> We truly appreciate your generous assessment of our method, particularly noting the highly novel and interesting core idea of self-selection, the strong generality, and its low barrier to adoption as a plug-and-play tool.
> We would like to address your concerns point by point below.
>
> **Q1.** The main weakness of this paper lies in its methodology, which is built on an assumption that may not always hold, and the solution relies to some extent on elaborate engineering tricks, which may affect its robustness and generality.
> Questionable Core Methodological Assumption: The core of SAVCD, the SAS prompt, relies on a critical assumption: that a model prone to hallucination and reasoning failures can accurately interpret a complex prompt to select an optimal corrective strategy for itself.
> This introduces a degree of circular reasoning. If the model's foundational reasoning is strong enough to perform this selection task perfectly, the initial problem of hallucination might be less severe. This design makes the method's success partially dependent on the very problem it aims to solve.
>
> **A1.** Thank you for your insightful comment.
> We do expect that models can select proper augmentations because the task that exposes the model weakness to hallucinate in a multimodal context is a different and more complex skill than the one required for the SAS prompt.
> **SAS is a text-only, meta-level classification task that simply requires the model to understand the operational knowledge of visual augmentations**.
> For example, answering a text-only question which augmentation is optimal for a text query "Which direction is the man facing?" with given operational instructions is completely different from answering the exact direction the man is facing with a given image.
> Therefore, SAS is a simpler task we can use to guide the model, which is still instruction-tuned with the large-scale text corpus, even if it is weak.
>
> We acknowledge that the model robustness may affect the effectiveness of our method.
> However, **we observe that the InstructBLIP model has the biggest relative performance gain from our method while exhibiting the weakest benchmark score overall**.
> This confirms that models with less effective foundational reasoning and lower benchmark scores benefit more from the targeted correction provided by SAVCD.
> **This demonstrates that SAVCD's success is not completely contingent on the inherent strength of the model**.
> Instead, it proves that the method effectively utilizes an **intact, general skill (Task-based reasoning) to correct for a highly complex, flawed skill (multimodal perception), thus avoiding any degree of circular dependence**.

---

> ### Author Response · Authors · 2025-11-20
>
> **Q2.** Over-reliance on Prompt Engineering: The method's effectiveness is heavily dependent on a meticulously crafted, complex prompt that includes extensive descriptions and examples. Such approaches are often brittle, being sensitive to changes in wording, structure, or even model versions. Furthermore, its scalability is limited; for instance, introducing new augmentation types would require manually modifying and debugging this complex prompt. This makes the method feel more like a highly-tuned heuristic for a specific set of tasks and models rather than a general, robust principle.
>
> **A2.** We appreciate this comment as it allows us to clarify the distinction between our SAS method and the specific prompt implementation used to deploy it.
> We agree that methods relying on a specific prompt are brittle [1].
> However, SAS is not a specific prompt, but rather a **generalizable concept** for a meta-level task.
> This is analogous to Chain-of-Thought [2], which is not one specific prompt, but a **general idea** of instructing a model to decompose a complex task into intermediate reasoning and sub-tasks.
> Likewise, SAS Prompting is beyond prompt engineering, a **concept** that leverages model knowledge to select an optimal visual counterpart.
>
> There are two pieces of evidence to support this conceptual framing.
> First, SAVCD is highly scalable. If we ought to introduce new augmentations to the set, the only necessary thing we should provide is a simple operational instruction of the new augmentation.
> We do not need to craft another in-context example, since prior studies have revealed that simply increasing the number of in-context examples does not always guarantee a performance increase, and the gain saturates as the example scales [3, 4].
>
> Second, SAVCD shows lower latency when configured with simple prompting, as shown in Table 3 in our main script.
> With the most minimal setting with only operational instructions (denoted as SAVCD-), it exhibits lower latency while minimal degradation of performance.
> We refer to the exact prompt of SAVCD- to the bottom attachment of your Q5 (Listing 1).
> Below, the updated Table 3 is attached to further show the averaged score on the benchmark.
>
> | Decoding | Scale | Num Tokens | Latency | Throughput | Avg. Score |
> | :--- | :---: | :---: | :---: | :---: | :---: |
> | VCD | 7B | 9914 | 18.50 | 54.06 | $69.08_{\pm 2.07}$ |
> | VCD | 13B | 8785 | 14.01 | 71.38 | $73.62_{\pm 1.40}$ |
> | VACoDe | 7B | 8418 | 16.97 | 58.93 | $69.12_{\pm 2.48}$ |
> | VACoDe | 13B | 8568 | 13.03 | 76.76 | $74.56_{\pm 1.62}$ |
> | SAVCD- | 7B | 8346 | 17.39 | 57.50 | $69.20_{\pm 2.12}$ |
> | SAVCD- | 13B | 8793 | 11.37 | 87.92 | $73.82_{\pm 1.65}$ |
> | SAVCD+ | 7B | 10163 | 15.08 | 66.32 | $69.22_{\pm 1.80}$ |
> | SAVCD+ | 13B | 8805 | 11.33 | 88.26 | $76.24_{\pm 0.83}$ |
>
> **Table 1.** Updated Tab. 3 in our main script. We added average and std for the scores.
>
> Therefore, we argue that SAVCD is a robust and scalable concept.
> It is more scalable than methods like VACoDe due to the simple addition of new augmentations, and provides a controllable trade-off between latency and performance.
>
> [1] Parameswaran et al. Revisiting Prompt Engineering via Declarative Crowdsourcing. arXiv preprint arXiv:2308.03854
>
> [2] Wei et al. Chain-of-thought prompting elicits reasoning in large language models. NeurIPS 2022
>
> [3] Zhang et al. Automated root causing of cloud incidents using in-context learning with gpt-4. SIGSOFT FSE Companion 2024
>
> [4] Nguyen et al. In-context eample selection with influences. arXiv preprint arXiv:2302.11042v2

---

> ### Author Response · Authors · 2025-11-20
>
> **Q3.** Regarding the robustness of SAS selection: What is the accuracy of the model's augmentation selection in the first step? Are there cases where the model selects a suboptimal or incorrect augmentation, leading to worse results than the baseline? Could the authors provide an analysis of such failure cases?
>
> **A3.** Thank you for your question.
> SAS is a **new conceptual task** therefore, ground truth does not exist, and human annotation would be subjective and potentially biased.
> To address this, we first input the SAS Prompt to a proprietary model (GPT-4o-mini) to generate a set of optimal augmentation choices.
> We refer to it as the "Oracle", assuming it to be the ground truth.
> The results for LLaVA-1.5-7B and 13B against Oracle are measured in terms of choice quality and LLM-as-a-Judge reasoning quality on lines 402, 405, 1134, and Table 7.
> The results reveal that **larger models are capable of generating better reasoning chains and choices**; The 7B model achieved an agreement score of 64.15\% and a reasoning quality score of 8.28, while the 13B model achieved an agreement score of 66.19\% and a reasoning quality score of 9.04.
>
> In addition, to address the concern about suboptimal selection, we refer to the "Static" strategy results in Table 4.
> These rows represent a scenario where the model is forced to use a single, highly likely to be suboptimal augmentation for all queries.
> Even in these non-favorable selection cases, **the worst performance (only random-mask, 1302.50) is still significantly higher than the baseline (1278.42)**.
>
> ---
>
> **Q4.** Regarding the limitations of the SAT algorithm: The core of SAT is using entropy to measure model uncertainty. However, models can sometimes produce a completely wrong hallucination with very high confidence (i.e., very low entropy). In such "confidently wrong" scenarios, could SAT's stricter threshold inadvertently lock in this error, making it harder to correct?
>
> **A4.** Thank you for this question that allows us to discuss about failure cases.
> We also recognize that a model confidently producing hallucinations is a known failure mode for language models.
>
> The failure case "confidently wrong" is what contrastive decoding-based methods are trying to mitigate.
> Suppose that we have a false positive index $i$ in both expert logit $l$ and amateur logit $l'$.
> It is assumed that $l'$ has a higher likelihood on $i$-th index ($l[i] < l'[i]$) due to the disruption from visual augmentations.
> Then, the index $i$ of contrasted logit $l_{\rm CD}$ value will decrease because $(1 + \alpha) \cdot l[i] - \alpha \cdot l'[i] < l[i]$ always holds by the assumption $l[i] < l'[i]$.
> Therefore, to mitigate such false positives, an example of this case is shown in the left side of Figure 2, denoting this with failure correction.
>
> Although the assumption is valid, SAT can still cut off the $i$-th logit value.
> There are several reasons why this cannot be the case.
> First, the design of the SAT prevents this by ensuring the threshold is **strictly less than 1** due to the range of the sigmoid function.
> This prevents the thresholding from pruning all the other tokens in the vocab set, as mentioned on line 236 of the main script.
> Second, our observations on experimental results reveal that models exhibit **high confidence in easy tokens** (e.g., articles, prepositions, punctuations).
> This empirical finding corroborates with a prior work [2].
> Based on this, we believe that such hallucinations with very high confidence are not likely to happen at the first place.
> Third, "lock-in" scenario can happen if SAT is used **in isolation** without any contrastive subtractions.
> However, SAT is applied after refining the logit distribution, rewarding the $i$-th logit value can avoid "lock-in" scenarios.
>
> Therefore, the failure case in all contrastive decoding-based methods would be **the case when the assumption for contrastive decoding is invalidated** ($l[i] > l'[i]$).
> That's why all CD-based methods try to find maximally contrastive counterparts.
> Determining and addressing such failure is beyond the scope of this study.
>
> [1] Li et al. Contrastive decoding: open-ended text generation as optimization. ACL 2023.
>
> [2] Chuang et al. Dola: decoding by contrasting layers improves factuality in large language models. ICLR 2024

---

> ### Author Response · Authors · 2025-11-20
>
> **Q5.** Regarding the generality of the prompt: The SAS prompt appears quite elaborate. How much would the performance degrade if this prompt were simplified (e.g., providing only the names and simple descriptions of augmentations, without the ICL examples)? This would help in understanding to what extent the performance gain comes from the method itself versus the elaborate prompt engineering.
>
> **A5.** Thank you for your question.
> SAS Prompt consists of three key components: provision of definitions of each augmentation, instruction to elicit reasoning, and in-context examples.
> We hypothesize that **the only necessary component is the provision of operational knowledge.**
> Table 6 shows the results on ablated configurations of reasoning instruction and in-context examples.
> The simplest configuration without both components results in a performance degradation of less than 1 percent of when compared to the full configuration, and 11 percent stronger than regular sampling.
> This implies that the performance gain from SAVCD does not originate from the design of the prompt, but rather its core concept of query-augmentation coupling.
>
> To further validate this, we ran the "name-only" prompt as suggested.
> Specifically, we instructed the following prompt to LLaVA-1.5-7B as:
> "Select the one image augmentation from the list that most invalidates the premise of the given question.\textbackslash n Options: [vertical flip, color inversion, random crop, random mask, random noise, horizontal flip]\textbackslash n Question: {}"
> This configuration removed all components of SAS Prompt including operational instructions, yet it scored 1312.39$_{\pm 11.71}$ on MME-Perception. This result is still better than the sampling baseline (1278.42) but falls within the range of the suboptimal "Static" configurations.
> This confirms our hypothesis: the ICL and reasoning are **optional**, but the operational knowledge is a necessary component.
> Therefore, we argue that SAS is a robust and scalable concept, superior to the other adaptive visual contrastive decoding approaches due to its simple scalability and controllable performance/latency trade-off.
>
> | Configuration | MME-P |
> | :--- | --- |
> | Baseline | $1278.42_{\pm 30.30}$ |
> | Name-only | $1312.39_{\pm 11.71}$ |
> | Only-OK | $1419.08_{\pm 11.39}$ |
> | OK + reasoning | $1428.02_{\pm 7.92}$ |
> | OK + ICL | $1428.63_{\pm 31.85}$ |
> | Full-prompting | $1431.30_{\pm 13.8}$ |
>
> **Table 2.** Results by prompt configurations. We denote OK for operational knowledge for this table.
>
> > You are an expert data augmentation analyst. Your task is to select the single most semantically disruptive image augmentation that most effectively invalidates the question's premise or prevents a confident answer.
> >
> > \## Augmentations and Their Effects ##
> > - Vertical flip: Flips image top-to-bottom. Disrupts questions about “above”, “below”, “under” or reading orientation.
> > - Color inversion: Replaces each color with its complement. Disrupts questions relying on accurate color identification.
> > - Random crop: Removes random parts of the image. Disrupts questions requiring global context or peripheral objects.
> > - Random mask: Occludes portions of the image. Disrupts object presence, count, or attribute recognition.
> > - Noise: Adds visual distortion. Disrupts questions requiring small details, texture, or text clarity.
> > - Horizontal flip: Flips the image left-to-right. Disrupts questions about left/right positioning and left-to-right text reading.
> >
> > \## Your Answer ##
> > If multiple augmentations could disrupt the question, select the one whose effect is most direct and unambiguous. You must choose one of the given augmentations following the "Choice:" format.
> >
> > Question: "{}"
>
> **Listing 1.** The exact prompted used for SAVCD-. This configuration only provides operational instructions for choosing the most optimal visual augmentation for the given multimodal query.

---

> ### Comment · Reviewer_msRD · 2025-11-28
> **Reply to rebuttal**
>
> Thank you for the author's careful response. I believe this paper presents a relatively good plug-and-play method. It would be great if the additional experiments and analyses could be included in the paper. I am willing to raise the score to 6.

---

> > ### Author Response · Authors · 2025-11-28
> >
> > We sincerely thank the reviewer for their thoughtful engagement throughout the review process and raising the score. We greatly appreciate the constructive feedback and insightful comments regarding a deeper investigation into the underlying mechanisms and potential solutions. We will bear these in mind when revising our paper for future publication. Thank you once again for your time and valuable guidance.

---

### Official Review · Reviewer_1xaB · 2025-11-01

**Soundness:** 2
**Presentation:** 2
**Contribution:** 2
**Rating:** 4
**Confidence:** 5

**Summary:**

This paper proposes Self-augmented Visual Contrastive Decoding, a training-free decoding strategy designed to mitigate hallucination problem in Large Vision-Language Models (LVLMs). SAVCD consists of two main components. First, it dynamically selects a query specific visual augmentation that produces maximally informative discrepancies between expert and amateur logits. Second, it applies contrastive decoding based on the self-augmented selection-based augmented image to penalize tokens that are likely to cause hallucination, without requiring additional training. The method significantly reduces hallucinations compared to existing baselines and achieves high computational efficiency

**Strengths:**

Strengths are summarized as follows:

(1) Conceptual Advancement: Query-Aware model-driven self-augmentation to reduce the hallucination effect of VLMs.

(2) Practical and Efficient mitigation without retraining method.

(3) Comprehensive evaluation and well-written to follow.

**Weaknesses:**

(1) The performance of SAVCD remains dependent on the underlying LVLM, limiting its robustness when transferred to different architectures or weaker base models. It would strengthen the paper to include expedriments on multiple LVLMs with varying parmeter scales to show cross-model consistency.

(2) The range of available visual augmentation is still constrained to a predfined candidate set, even though the method dynamically selects among them.

(3) VaCoDe Misrepresentation; While the paper highlights a major distinction fro mVaCoDe, the description partially misrepresents VaCoDe's mechanism. Specifically, VaCoDe also selects augmentations dynamically from a fixed set based on output divergence, rather than using purely fixed perturbations. It would strengthen that clarifying the comparison in the main text between VaCoDe and SAVCD.

**Questions:**

Please refer to the Weaknesses part.

---

> ### Author Response · Authors · 2025-11-20
>
> We truly appreciate your comments and finding our paper to be conceptually advantageous, practical and efficient, and well-written. We would like to address your concerns point by point below.
>
> ---
>
> **Q1.** The performance of SAVCD remains dependent on the underlying LVLM, limiting its robustness when transferred to different architectures or weaker base models. It would strengthen the paper to include experiments on multiple LVLMs with varying parameter scales to show cross-model consistency.
>
> **A1.** Thank you for your suggestion.
> We anticipate that most of the LVLMs can benefit from our decoding method since they share similar underlying structures, pre/post-training mechanisms across the model.
> To further empirically verify this claim and demonstrate the broad applicability of our method, we conducted additional experiments on the Qwen3-VL-8B and 32B, which represent a state-of-the-art LVLM family.
> Due to time and resource constraints, we report results 5 runs on MMVP, and LLaVA-Bench.
> SAVCD demonstrate its generalizability by consistently outperforming the other decoding methods on newer models, as shown below.
> We will add the full experimental results on the new version of our manuscript.
>
> | Model | Decoding | MMVP | LLaVA-Bench |
> | :--- | :--- | :---: | :---: |
> | Qwen3-VL-8B | Multinomial | $55.87_{\pm 1.19}$ | $120.42_{\pm 1.90}$ |
> | Qwen3-VL-8B | VCD | $56.27_{\pm 1.80}$ | $120.40_{\pm 1.36}$ |
> | Qwen3-VL-8B | VACoDe | $55.20_{\pm 0.56}$ | $120.60_{\pm 0.99}$ |
> | Qwen3-VL-8B | SAVCD | $57.87_{\pm 1.59}$ | $121.84_{\pm 0.75}$ |
> | Qwen3-VL-32B | Multinomial | $53.73_{\pm 1.19}$ | $122.66_{\pm 1.88}$ |
> | Qwen3-VL-32B | VCD | $55.87_{\pm 0.56}$ | $122.54_{\pm 1.13}$ |
> | Qwen3-VL-32B | VACoDe | $52.00_{\pm 1.94}$ | $121.34_{\pm 1.74}$ |
> | Qwen3-VL-32B | SAVCD | $57.20_{\pm 2.13}$ | $123.18_{\pm 1.17}$ |
>
> **Table 1.** Results with Qwen3-VL on MMVP and LLaVA-Bench.
>
> ---
>
> **Q2.** The range of available visual augmentation is still constrained to a predefined candidate set, even though the method dynamically selects among them.
>
> **A2.** Thank you for your comment.
> We acknowledge that the predefined set of six augmentations is a limitation, as mentioned in the "Limitations and Future Work" section.
> This set was adopted from the established methodology in prior work [1], which has already shown that these specific transformations are effective at targeting a diverse range of visual capabilities.
> This set is sufficiently diverse because random noise hinders holistic and compositional understanding, both flips can invalidate any spatial-related queries, color and appearance disrupt attributal perception, and random masking and cropping can be applied broadly since they occlude the object-of-interest and background information, respectively.
> We would like to further demonstrate how the augmentation set has wide coverage and is sufficiently diverse using nine different visual patterns identified in the MMVP [2] paper as an example.
>
> - Color and Appearance: Challenged by color inversion, which forces the model to identify objects by attributes and context rather than relying on learned color properties.
> - Orient&Direction, Positional&Relational Context, Viewpoint&Perspective: Directly invalidated by horizontal and vertical flip.
> - Text (OCR), State&Condition, and Structural Characteristics: Random noise impedes holistic understanding of the scene.
> - Presence of Specific Features and Quantity&Count: Hindered by occlusion-based operations (random mask and crop).
>
> These coverages demonstrate that, despite its compact size, the augmentation pool covers a wide spectrum of visual stressors relevant to multimodal understanding. Extending this pool beyond a predefined set remains an important direction for future work, and we intend to explore more adaptive and data-driven augmentation mechanisms as part of our subsequent research.
>
> [1] Kim et al. Vacode: Visual augmented contrastive decoding. arXiv preprint arXiv:2408.05337
>
> [2] Tong et al. Eyes wide shut? exploring the visual shortcomings of multimodal llms. CVPR 2024

---

> ### Author Response · Authors · 2025-11-20
>
> **Q3.** VaCoDe Misrepresentation; While the paper highlights a major distinction from VaCoDe, the description partially misrepresents VaCoDe's mechanism. Specifically, VaCoDe also selects augmentations dynamically from a fixed set based on output divergence, rather than using purely fixed perturbations. It would strengthen that clarifying the comparison in the main text between VaCoDe and SAVCD.
>
> **A3.** Thank you for your comment.
> We would like to clarify that **we also recognize VACoDe as a pioneering work of dynamic methods** for contrastive decoding in a multimodal context, as mentioned on lines 53, 129, 362, and Table 4.
> However, we identify three limitations of VACoDe.
> (1) Computational Bottleneck: VACoDe relies on a brute-force mechanism to measure divergence for every candidate augmentation, which is computationally expensive.
> (2) Dependence on First Token: Its selection is overly dependent on the divergence observed at the very first generated token, potentially missing crucial context for longer sequences.
> (3) Benchmark Brittleness: In practice, VACoDe was required to manually change its set of augmentations based on the evaluation benchmark, confirming the fragility of its heuristic approach.
>
> The main purpose of SAS Prompting is **addressing the limitations of VACoDe by using a single, universal augmentation set across all benchmarks and employing a meta-level task to directly couple the query intent and visual derivatives**.
> This ensures our system is principled, controllable, and selects the correct visual derivative without the brute-force cost or dependence on premature divergence criteria.

---

### Official Review · Reviewer_cNnA · 2025-11-03

**Soundness:** 3
**Presentation:** 3
**Contribution:** 2
**Rating:** 6
**Confidence:** 2

**Summary:**

This paper addresses the issue of hallucination in Large Vision-Language Models (LVLMs) to improve their generation quality. Previous works have employed contrastive decoding (CD) to enhance the output of Large Language Models (LLMs), an approach later extended to the visual domain as Visual Contrastive Decoding (VCD). Although promising, existing VCD methods often rely on generic visual augmentations that are agnostic to the specific context of the text query, limiting their effectiveness. To address this, this paper introduces a training-free decoding strategy with two key innovations: first, a self-augmentation prompting technique that leverages the model's intrinsic knowledge to dynamically align semantics between the query and visual content; and second, an adaptive thresholding algorithm that adjusts the candidate token set based on output sparsity, making full use of the logit distribution. Extensive experiments validate the method's effectiveness across four LVLMs and seven benchmarks.

**Strengths:**

This paper is well-organized with good writting. Importantly, the authors provide extensive experiments to validate the effectiveness of their method.

**Weaknesses:**

**Novelty Assessment**: As I am not deeply familiar with the contrastive decoding literature, I cannot authoritatively assess the novelty of the proposed method against all prior works in Visual Contrastive Decoding (VCD). The idea appears novel based on my reading, but a more thorough discussion of related work would help situate this contribution for a broader audience. I look forward to the discussion with other reviewers and the authors' response on this point.

**Model Generalization**: The experimental validation, while extensive, relies on a set of LVLMs that does not include several recent state-of-the-art models (e.g., InternVL3/3.5, Qwen-VL2/3). To strengthen the claim of generalizability, it would be valuable to see if the proposed SAVCD method's benefits hold on these newer architectures.

Note to the AC and Reviewers: Given my limited background in this specific sub-field, I will rely heavily on the discussion and author response to finalize my assessment, and I defer to the experts on the finer points of the related work.

**Questions:**

Please refer to my weakness

---

> ### Author Response · Authors · 2025-11-20
>
> We truly appreciate your comments and finding our paper to be well-written and methods to be effective. We would like to address your concerns point by point below.
>
> ---
>
> **Q1.** Novelty Assessment: As I am not deeply familiar with the contrastive decoding literature, I cannot authoritatively assess the novelty of the proposed method against all prior works in Visual Contrastive Decoding (VCD). The idea appears novel based on my reading, but a more thorough discussion of related work would help situate this contribution for a broader audience.
>
> **A1.** Thank you for your comment.
> We would like to clarify the distinct novelty of our two key contributions.
>
> First, VCD [1] and following works [2,3] focus on finding the contrastive image counterpart by modifying the input image in **identical underlying heuristics**.
> Even though the counterpart varies by factors such as randomness (e.g., random noise), the main functionality of the algorithms remains **static** and query-agnostic, failing to adapt to the different cognitive tasks required by diverse queries.
> While VACoDe [4] also proposed a dynamic visual augmentation strategy, new limitations on its brute-force mechanism and dependence on the first token emerged as mentioned in lines 55 and 132.
> Also, they use different augmentation sets by benchmark in practice.
> SAS Prompt is proposed to address the limitations of dynamic VCD in the current state by **coupling the query intent and visual derivatives**, with more controllable and flexible computational overhead.
>
> Separately, this work improves APC to SAT.
> Contrastive decoding may undesirably reward false negatives; APC filters them by setting a threshold, which is determined by subtracting a **fixed** parameter from the max logit value.
> However, static thresholding may result in improper pruning depending on the overall sparsity of the logit since **each logit value is relative to each other in the distribution**.
> SAT mitigates this limitation by leveraging entropy, a relational information between the logit values within the distribution.
> This enables SAT to calculate a dynamic threshold that correctly adapts its leniency based on the model's objective state of uncertainty, avoiding the improper pruning that plagues static, confidence-agnostic filtering of APC.
>
> We will further clarify these in the camera-ready version of our manuscript to highlight novelty.
>
> [1] Leng et al. Mitigating object hallucinations in large vision-language models through visual contrastive decoding. CVPR 2024
>
> [2] Huang et al. Opera: Alleviating hallucination in multi-modal large language models
> via over-trust penalty and retrospection-allocation. CVPR 2024
>
> [3] Wang et al. Mitigating hallucinations in large vision-language models with instruction contrastive decoding. ACL 2024
>
> [4] Kim et al. Vacode: Visual augmented contrastive decoding. arXiv preprint arXiv:2408.05337
>
> ---
>
> **Q2.** Model Generalization: The experimental validation, while extensive, relies on a set of LVLMs that does not include several recent state-of-the-art models (e.g., InternVL3/3.5, Qwen-VL2/3). To strengthen the claim of generalizability, it would be valuable to see if the proposed SAVCD method's benefits hold on these newer architectures.
>
> **A2.** Thank you for your suggestion.
> We anticipate that most of the LVLMs can benefit from our decoding method since they share similar underlying structures, pre/post-training mechanisms across the model.
> To further empirically verify this claim and to demonstrate the generalizability and robustness of our method, we conducted additional experiments on the Qwen3-VL-8B and 32B, which represent a state-of-the-art LVLM family.
> Due to time and computational resource constraints, we report results from 5 runs on MMVP and LLaVA-Bench.
> SAVCD demonstrate its generalizability by consistently outperforming the other decoding methods on newer models, as shown below.
> We will add further experimental results with these models in the camera-ready version of our manuscript.
>
> | Model | Decoding | MMVP | LLaVA-Bench |
> | :--- | :--- | :---: | :---: |
> | Qwen3-VL-8B | Multinomial | $55.87_{\pm 1.19}$ | $120.42_{\pm 1.90}$ |
> | Qwen3-VL-8B | VCD | $56.27_{\pm 1.80}$ | $120.40_{\pm 1.36}$ |
> | Qwen3-VL-8B | VACoDe | $55.20_{\pm 0.56}$ | $120.60_{\pm 0.99}$ |
> | Qwen3-VL-8B | SAVCD | $57.87_{\pm 1.59}$ | $121.84_{\pm 0.75}$ |
> | Qwen3-VL-32B | Multinomial | $53.73_{\pm 1.19}$ | $122.66_{\pm 1.88}$ |
> | Qwen3-VL-32B | VCD | $55.87_{\pm 0.56}$ | $122.54_{\pm 1.13}$ |
> | Qwen3-VL-32B | VACoDe | $52.00_{\pm 1.94}$ | $121.34_{\pm 1.74}$ |
> | Qwen3-VL-32B | SAVCD | $57.20_{\pm 2.13}$ | $123.18_{\pm 1.17}$ |
>
> **Table 1.** Results with Qwen3-VL on MMVP and LLaVA-Bench.

---

### Author Response · Authors · 2025-12-02

We sincerely appreciate the constructive feedback and are highly encouraged by the reviewers' positive assessment of our work.
## 1. Significance and Impact
Our line of work emphasizes the importance of the semantic coupling for query-augmentation and confidence-sensitive decoding as a principled approach for developing more robust multimodal generation. By providing that general knowledge can guide specific visual execution, we advance the generality of reliable multimodal decoding strategies while keeping the computational cost controllable.

## 2. Recognition of Strengths
The reviewers consistently recognize SAVCD as a significant contribution to the field. They specifically commented:
- Conceptual Novelty: The core idea of query-aware visual augmentation (SAS) is highly novel, interesting, and offering a new perspective on mitigating LVLM hallucinations (Reviewer 1xaB, msRD).
- Practical Utility: The method is recognized as a practical and efficient inference-time strategy that grants strong generality and easy adoption with intuitive explanations (Reviewer 1xaB, msRD, poDY)
- Thorough Evaluation: The extensive experiments covering 4 LVLMs across 7 benchmarks demonstrate a comprehensive evaluation and a well-written, easy-to-follow manuscript (All Reviewers).
## 3. Enhancements during Rebuttal
To address specific concerns and further strengthen the manuscript, we conduct extensive additional analyses:
- Generalizability to newer models: We perform additional experiments on newer model, Qwen3-VL-8B and 32B and found that SAVCD benefit on better performing contemporary models (Reviewer cNnA, 1xaB, poDY)
- Prompt insensitivity: SAS Prompting is effective if the operational knowledge is provided, demonstrating the generalizable concept of finding amateur counterpart (Reviewer msRD).
- Task Novelty: SAS is a meta-level classification task that effectively utilizes an intact, general skill (Task-based reasoning) to correct for a highly complex, flawed skill (multimodal perception), avoiding any degree of circular dependence.
## 4. Conclusion
Following these clarifications and additional experiments, Reviewer msRD increased the score from 4 to 6, and Reviewer poDY affirmed positive ratings. In the future version our manuscript will include the changes:
- Include full results on Qwen3-VL model families, ablations on minimal-operation knowledge and name-only configurations, and clarify the novelty
- In addition, after our internal discussion, we will modify the paper title to "Self-Aug: Query and Sparsity Aware Decoding for Large Vision Language Model Hallucinations" to further highlight the contribution of SAT.

---

### Meta-Review · Area_Chair_VFCt · 2026-01-06

**Summary:**

Reviewers’ key concerns: Incremental novelty vs VCD/VACoDe, generalization to newer models (e.g., Qwen3-VL), SAS’s predefined augmentations/prompt dependency, circular reasoning in SAS, and experimental robustness, etc. These were addressed from the rebuttal.

**Reviewer Concerns:**

Addressed concerns include: Model generalization, cross-model consistency, arbitrary augmentation set, etc.

No outstanding concerns.

**Reviewer Scores:**

Reviewer msRD could possiblly increase the score if additional experiments can be added in the publication version.

---

### Decision · Program_Chairs · 2026-01-26

Accept (Poster)